# CAR-T cell therapy-related cytokine release syndrome and therapeutic response is modulated by the gut microbiome in hematologic malignancies

Yongxian Hu[1,2,3,4,12], Jingjing Li [5,6,12], Fang Ni[1,2,3,4,12], Zhongli Yang[5,6,12], Xiaohua Gui[5,6], Zhiwei Bao[5,6], Houli Zhao[1,2,3,4], Guoqing Wei[1,2,3,4], Yiyun Wang[1,2,3,4], Mingming Zhang[1,2,3,4], Ruimin Hong[1,2,3,4], Linqin Wang[1,2,3,4], Wenjun Wu[1,2,3,4], Mohamad Mohty[7,8], Arnon Nagler[9], Alex H. Chang [10] ✉, Marcel R. M. van den Brink [11] ✉, Ming D. Li[5,6] ✉ & He Huang [1,2,3,4] ✉

Immunotherapy utilizing chimeric antigen receptor T cell (CAR-T) therapy holds promise for hematologic malignancies, however, response rates and associated immune-related adverse effects widely vary among patients. Here we show, by comparing diversity and composition of the gut microbiome during different CAR-T therapeutic phases in the clinical trial ChiCTR1800017404, that the gut flora characteristically differs among patients and according to treatment stages, and might also reflect patient response to therapy in relapsed/refractory multiple myeloma (MM; $n = 43$), acute lympholastic leukemia (ALL; $n = 23$) and non-Hodgkin lymphoma (NHL; $n = 12$). We observe significant temporal differences in diversity and abundance of Bifidobacterium, Prevotella, Sutterella, and Collinsella between MM patients in complete remission ($n = 24$) and those in partial remission ($n = 11$). Furthermore, we find that patients with severe cytokine release syndrome present with higher abundance of Bifidobacterium, Leuconostoc, Stenotrophomonas, and Staphylococcus, which is reproducible in an independent cohort of 38 MM patients. This study has important implications for understanding the biological role of the microbiome in CAR-T treatment responsiveness of hematologic malignancy patients, and may guide therapeutic intervention to increase efficacy. The success rate of CAR-T cell therapy is high in blood cancers, yet individual patient characteristics might reduce therapeutic benefit. Here we show that therapeutic response in MM, ALL and NHL, and occurrence of severe cytokine release syndrome in multiple myeloma are associated with specific gut microbiome alterations.

B-cell-derived hematologic malignancies, including acute lymphoblastic leukemia (B-ALL), non-Hodgkin lymphoma (B-NHL), and multiple myeloma (MM), carry a high probability of relapse after conventional chemotherapy[1]. With novel therapeutic strategies incorporating monoclonal antibodies, bispecific T-cell engager (BiTE) antibodies, and hematopoietic stem cell transplantation (HSCT), treatment outcomes have greatly improved[2–4]. However, some patients progress to relapsed/refractory (r/r) status, with a poor prognosis[5]. For patients with r/r B-ALL, the median overall survival (OS) is 3–6 months[6, 7]. The median OS is 6.2 months for patients with r/r diffuse large B-cell lymphoma (DLBCL)[8]. For r/r MM patients, the median OS is 3–9 months[9]. There is an urgent need to explore novel treatment strategies for these malignancies.

Chimeric antigen receptor (CAR) T-cell therapy (approved by the U.S. Food and Drug Administration) recently emerged as promising for r/r B-ALL, DLBCL, and mantle cell lymphoma (MCL)[10–12]. In multiple myeloma, investigations targeting the B-cell maturation antigen (BCMA) yielded encouraging outcomes with reversible toxic effects such as cytokine release syndrome (CRS) and pancytopenia[13–17]. However, the efficacy and toxicity have been inconsistent. No biomarker has been identified that can predict outcomes and associated toxicities after CAR-T therapy in patients.

Several studies have reported that the differences in diversity and composition of the gut microbiome might influence cancer immunotherapy response[18–21]. After analyzing fecal samples from 43 melanoma patients treated with anti-programmed cell death 1 protein (PD-1) immunotherapy, significantly higher alpha diversity and abundance of Clostridiales/Ruminococcaceae were found in responders, whereas Bacteroidales were significantly enriched in non-responders[19]. In hematologic malignancies, intestinal bacteria also modulate the risk of graft-versus-host disease (GVHD) and infection after allogeneic hematopoietic stem cell transplantation (allo-HSCT). Greater bacterial diversity and abundance of the genus *Blautia* were associated with reduced GVHD-related death and improved OS[22, 23]. However, no study has shown a potential role for the intestinal microbiota in the efficacy and toxicity of CAR-T therapy for B-cell malignancies.

Here we show the intestinal microbiome changes in patients with r/r B-cell-derived hematologic malignancies undergoing CAR-T cell treatment and investigating associations of the microbiota with clinical responses and CRS severity. Further, the potential of the gut microbiome to predict treatment outcomes and CRS severity was also explored. Our results indicated that CAR-T cell therapy-related cytokine release syndrome and therapeutic response was modulated by the gut microbiome in hematologic malignancies. These findings highlight the role of gut microbiome in CAR-T therapy.

## Results
### Clinical trial outcomes
Previously we reported the safety and efficacy of interim results of the trial (61 patients)[24]. Here after completion of the trial, 99 patients with relapsed/refractory multiple myeloma (r/r MM) were included (Fig. 1a). The primary outcome was to evaluate the safety of BCMA CAR-T cells in the treatment of r/r MM. All patients were evaluated for safety analysis. Cytokine release syndrome (CRS) was observed in 97% (96/99) patients, including 50 (52.1%) patients with grades 1–2 CRS, 42 (43.8%) and 4 (4.1%) with grades 3 and 4 CRS. None grade 5 CRS occurred. The neurotoxicities were reported for 11 patients (11.1%), of whom 10 (10.1%) and 1 (1.0%) had grade 1 and grade 2 events, no grade 3 or higher neurotoxic effect was observed. After treatment, all episodes of CRS and neurotoxicity were resolved. The secondary outcome was to evaluate the efficacy and characterization of BCMA CAR-T cells in the treatment of r/r MM. Within 1 month after BCMA CAR-T cell infusion, 2 patients died of cerebral hemorrhage and 4 died of severe infections. Of the 95 evaluable patients, 91 (95.8%) had an overall response. In all, 55.8% (53/95), 15.8% (15/95), and 24.2% (23/95) of patients achieved a complete remission

(CR), very good partial response (VGPR), or partial response (PR), respectively. With a median follow-up time of 21.2 months (95% CI, 18.4–32.1), the median progression free survival (PFS) was 12.0 (95% CI, 8.1–15.7) months. The 1-year OS and PFS rates were 0.70 (95% CI, 0.61–0.80) and 0.48 (95% CI, 0.39–0.59), respectively. BCMA CAR-T cells expanded dramatically in vivo. The BCMA CAR-T/CD3$^+$ T-cell percentages in peripheral blood (PB) peaked on day 11 (range: 5–31) after CAR-T cell infusion. The median BCMA CAR-T/CD3$^+$ T-cell percentages was 81.95% (range: 6.07–97.30%).

### Patient samples included for gut microbiome analysis
Microbiome samples were not available from 12 patients and 16S sequencing depth was not sufficient for analysis on 6 patients. Finally, a total of 81 patients with r/r MM was included for gut microbiome analysis, which included 43 patients for experiment group and 38 patients for validation group (Fig. 1a). Number of samples collected, and sequencing depth were summarized in Supplementary Data 1–2. Clinical and sequencing information of patients used in the study are presented in Supplementary Table 3 and Supplementary Data 3.

The median age of the MM patients was 59 (range 39–75) years, and 55.8% were male (Table 1). The median number of prior lines of therapy was 4 (range 2–8), with all receiving proteasome inhibitor therapy and 95.3% immunomodulatory agents. At enrollment, 39.5% had received autologous stem cell transplantation, and 55.8% had extramedullary disease(s).

Three months after infusion of a median dose of $4.4 \times 10^6$/kg (range $1.2$–$6.9 \times 10^6$/kg) of BCMA CAR-T cells, 55.8%, 14%, and 25.5% of patients had a CR, VGPR, or PR, respectively. All 43 MM patients showed CRS, grade 1 in 8 patients (18.6%), grade 2 in 16 (37.2%), and grade 3 in 19 (44.2%). No higher grade was observed (Fig. 1d). The CRS was fully controlled and managed for all patients. Of these patients, 24 received only supportive care, 6 received supportive care plus tocilizumab treatment (IL-6 receptor-blocking monoclonal antibody), 10 received supportive care and corticosteroid treatment, and 3 received supportive care accompanied with tocilizumab and corticosteroids treatment. The antibiotics used before or during treatment were β-lactam (41 patients), Carbapenems (26 patients), Quinolone (26 patients), Aminoglycosides (1 patient), Macrolide (1 patient), Tetracyclines (4 patients), Cephalosporins (3 patients), and Glycopeptides (6 patients). Although we included age, gender, number of prior lines of therapy, CAR-T cell dose, autologous stem cell transplantation, antibiotic use before or during treatment as covariates into our analyses, no significant differences were observed among different efficacy groups or CRS grade groups (Supplementary Tables 1–2). Two patients died: one from sepsis caused by *Pseudomonas aeruginosa* and the other from intracranial hemorrhage (Fig. 1d). Both the BCMA CAR-T/CD3$^+$ T-cell percentages in peripheral blood (PB) and serum concentrations of interleukin (IL)−10 increased during CRS and differed significantly in the CR and PR groups (Fig. 1e). Patients' temperature and C-reactive protein (CRP), ferritin, and lactic dehydrogenase (LDH) concentrations were elevated, and IL-6 and IFN-γ concentrations were significantly different in grade 3 vs grade 1 CRS (Fig. 1f and Supplementary Fig. 1a–c). The serum immunoglobulins (IgG, IgA) and immunoglobulin κ and λ light chain concentrations decreased dramatically after CAR-T (Supplementary Fig. 1d–f). Figure 1g shows the differences of positron emission tomography–computed tomography (PET-CT) scans and plasma cells detected by Wright's stain of a bone marrow smear (43.5% vs. 0), as well as flow cytometry (68.9% vs. 0) of bone marrow cells before and after CAR-T infusion for a representative subject.

### Changes in the intestinal microbiome during CAR-T cell therapy
To detect changes in the gut microbiota during CAR-T therapy, we collected fecal samples from each patient at five times (FCa, FCb, CRSa, CRSb, and CRSc; Fig. 1c), where FCa denotes the baseline before chemotherapy; FCb after chemotherapy; CRSa after CAR T-cell infusion

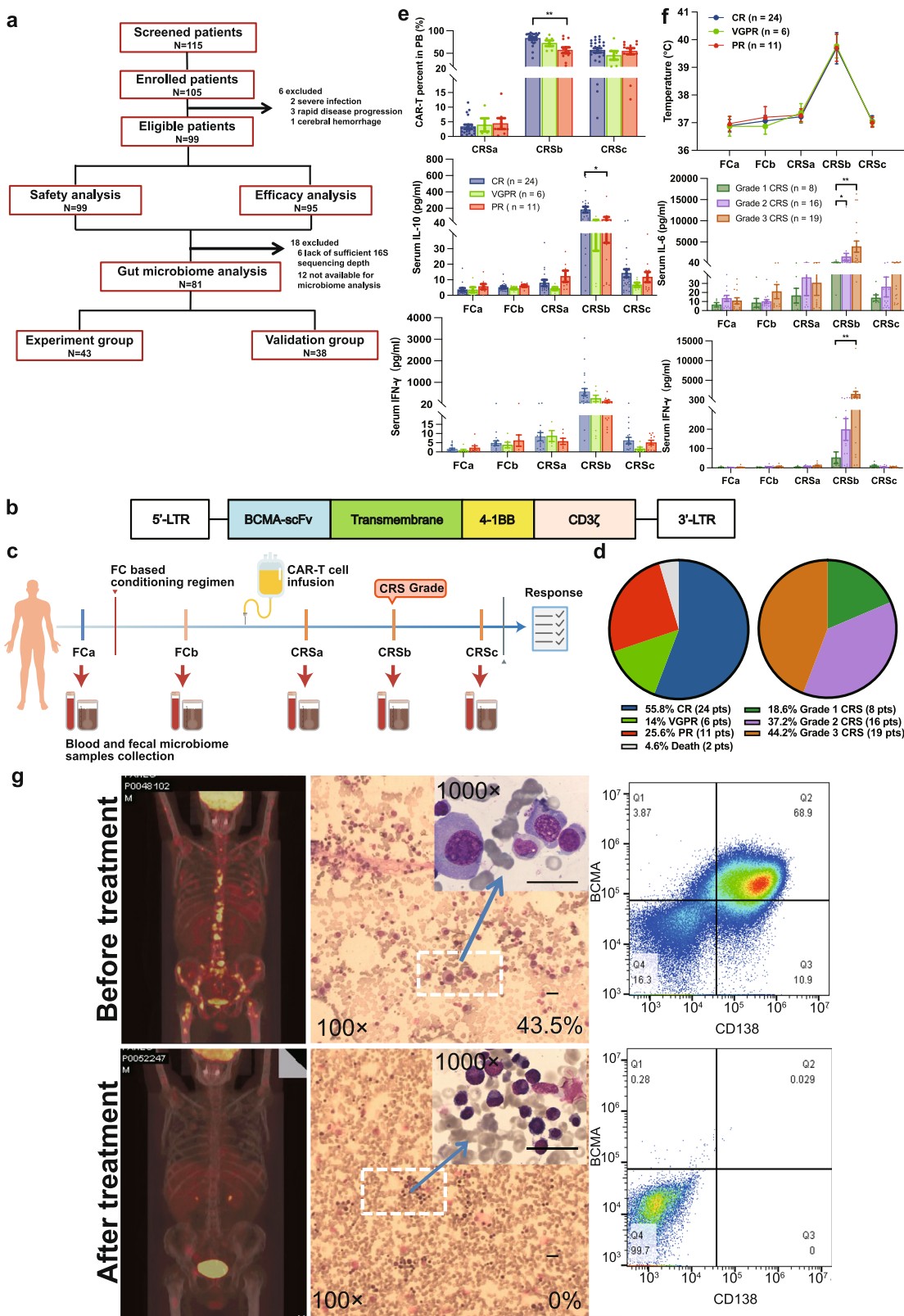

but before the onset of CRS; and CRSb and CRSc denote the peak and during the recovery phase of CRS, respectively. The median date of FCa was 4 days (range 2–7) before CAR-T cell infusion in MM patients, the median date of FCb was 0 days (range 0–7) before CAR-T cell infusion, and the median dates of CRSa, CRSb, CRSc after CAR-T cell infusion were 2 days (range 1–5.3), 6 days (range 2.5–17.4), and 14 days (range 8–37.5), respectively.

We first evaluated the diversity of the gut microbiota in all sub-jects during CAR-T cell therapy in MM patients. Compared with early stage, there was a significant decrease in diversity (measured by the Shannon index) after the CAR-T therapy (Fig. 2a). This decrease was observed in the microbiome of patients receiving CAR-T therapy for r/r ALL (Supplementary Fig. 4a) or r/r NHL (Supplementary Fig. 4b). Refer to Supplementary Table 3 for details on the characteristics of r/r B-ALL

**Fig. 1 | Trial profile and clinical response in r/r MM patients treated with CAR-T cell infusion. a** Patient enrollment. **b** Anti–BCMA single-chain variable fragment (scFv), a hinge and transmembrane regions, and 4-1BB costimulatory moiety, and CD3ζ T-cell activation domain. **c** Blood and fecal sample collection. **d** Clinical response; CRS grade distribution in 43 r/r MM patients. **e** Numbers of BCMA CAR-T cell percentages in PB assessed by FACS in different therapy stages after CAR-T cell infusion and serum concentrations of IL-10 and IFN-γ in different therapy stages among the CR (n = 24 biologically independent patients), VGPR (n = 6 biologically independent patients), and PR (n = 11 biologically independent patients) groups. Blue, green, and red colors indicate CR, VGPR, and PR group, respectively. Data are presented as mean values ± SEM. Significance determined by two-sided Kruskal-Wallis test and adjustments were made for multiple comparison. *P* values for CAR-T percent in PB, serum IL-10 and IFN-γ between CR and PR groups in CRSb stage were 0.004, 0.048, 0.085, respectively. *p < 0.05, **p < 0.01. **f** Body temperature and

serum concentrations of IL-6 and IFN-γ in different therapy stages among CRS grade groups. (Grade 1 CRS group: n = 8 biologically independent patients, Grade 2 CRS group: n = 16 biologically independent patients, and Grade 3 CRS group: n = 19 biologically independent patients). Data are presented as mean values ± SEM. Significance determined by two-sided Kruskal-Wallis test and adjustments were made for multiple comparison. *P* values for serum IL-6 and IFN-γ between Grade 1 CRS and Grade 3 CRS were 0.002 and 0.006, respectively. *p < 0.05, **p < 0.01. **g** Representative MM patients with impressive antimyeloma response. Positron emission tomography-computed tomography scans before and 5 months after CAR-T cell treatment showing complete elimination of large number of MM bone metastases. Before receiving CAR-T cell infusion, 43.5% of bone marrow cells of the patient were plasma cells, but after 1.5 months of infusion, dramatic eradication of MM from the bone marrow was observed; and MM cells became undetectable by flow cytometry. The bar indicates a length of 5 μm.

and B-NHL patients. In addition, we analyzed diversity change in an independent MM sample with 38 patients included and found a decreased Shannon index along different therapy stages (Supplementary Fig. 4c). To further assess the similarity of composition between different therapy stages, we performed pairwise Spearman correlation analysis of operational taxonomic unit (OTU) level bacterial abundance (Fig. 2b) and found that stronger correlations emerged during the early stages with a ρ value of 0.71, 0.73, and 0.68, respectively, at FCa, FCb, and CRSa. Correlations between late stages (CRSb and CRSc) and early stages were weaker, suggesting that changes in microbiome composition might be related to CRS.

We next explored community structure and temporal shift of bacterial abundance at multiple taxonomic levels during CAR-T therapy. In these myeloma patients, bacterial communities were dominated by Firmicutes and Bacteroidetes at the phylum level (Fig. 2c). Abundance

of Firmicutes increased but that of Bacteroidetes decreased at later stages compared with the baseline (Wilcoxon rank-sum test, p < 0.05, Supplementary Fig. 4d). By applying the longitudinal analysis in the Qiime2 microbiome analysis platform, we detected changes in the gut microbial communities at taxonomic levels from phylum to genus (Fig. 2d and Supplementary Data 3). We further employed a negative binominal (NB) regression model-based time-course analysis to identify genera with significant temporal changes (Supplementary Data 4). Five genera were detected by both Qiime2 and maSigPro procedures, which included increases in *Enterococcus*, *Lactobacillus*, and *Actinomyces* and decreases in *Bifidobacterium* and *Lachnospira* (bolded genera in Fig. 2d). Most changes were aggravated during the late stages (Supplementary Fig. 4e). Additionally, for repeated measure data (Subjects = 10), we applied Friedman's test and found nine genera affected significantly by CAR-T therapy among which the genus *Enterococcus* had the largest difference between stages (Fig. 2e).

Moreover, by checking changes in the five genera in ALL and NHL patients, we observed consistent shift trends in NHL (two genera; Supplementary Fig. 4f) and ALL (four genera; Supplementary Fig. 4g), respectively. These results were further verified in another independent MM sample, showing that CAR-T therapy correlated significantly with decreased Shannon diversity (Supplementary Fig. 4c) and increased abundance of genus *Enterococcus* and *Actinomyces* (Supplementary Fig. 4h).

## Association between microbial communities and clinical response to CAR-T therapy

We next determined whether microbial compositions or changes were associated with the response to CAR-T therapy. Because we wanted to identify maximum differences and only six subjects presented in the VGPR group, we performed comparisons only between the CR and PR groups.

In MM patients, notable differences in microbial alpha and within-sample diversity were observed in patients with CR and PR at CRSb stage (Fig. 3a, b). Although no differences were detected at baseline, PR patients descended more dramatically in alpha diversity and had significantly lower Shannon indices than CR patients after CAR-T infusion (Fig. 3a). As the degree of differences between CR and PR groups changed across therapeutic stages, we characterized the periods with greater differences by summarizing the amount of CR/PR-enriched OTU at each timepoint. The most pronounced differences occurred at CRSb (Fig. 3c).

To explore longitudinal differences between CR and PR across all therapeutic stages, we identified OTU features with differential dynamic profiles by applying negative binominal regression-based time-course differential analysis with the maSigPro package. In total, 125 OTUs were found to have differential time-course patterns between CR and PR patients (Fig. 3d and Supplementary Data 5). The significant OTUs were further grouped into three clusters according to profiles of their abundance. Most of these OTUs were in clusters 1 and 2 (Fig. 3e). Cluster 1, characterized by enrichment in the CR group, was comprised mainly

## Table 1 | Baseline characteristics of MM patients and validation MM patient samples included in final fecal microbiome analysis

| | Total N = 43 (%) | Validation sample N = 38 (%) |
|---|---|---|
| **Age** | | |
| Median | 59 | 60 |
| Range | 39–75 | 16-74 |
| **Gender** | | |
| Male | 24 (55.8) | 17 (44.7) |
| Female | 19 (44.2) | 21 (55.3) |
| **Number of prior lines of therapy** | | |
| Median | 4 | 4 |
| Range | 2–8 | 2–7 |
| **CAR-T cell dose (×106/kg)** | | |
| Median | 4.4 | 2.1 |
| Range | 1.2–6.9 | 0.74-6 |
| **Autologous stem cell transplantation** | | |
| No | 26 (60.5) | 25 (65.8) |
| Yes | 17 (39.5) | 13 (34.2) |
| **Extramedullary disease** | | |
| No | 19 (44.2) | 17 (44.7) |
| Yes | 24 (55.8) | 21 (55.3) |
| **Prior PI therapy** | | |
| No | 0 | 0 |
| Yes | 43 (100) | 38 (100) |
| **Prior IMiD therapy** | | |
| No | 2 (4.7) | 2 (5.3) |
| Yes | 41 (95.3) | 36 (94.7) |

*PI* Proteasome inhibitors (Bortezomib/Carfilzomib/Ixazomib), *IMiD* immunomodulatory agent (Lenalidomid/Thalidomid/Pomalidomide).

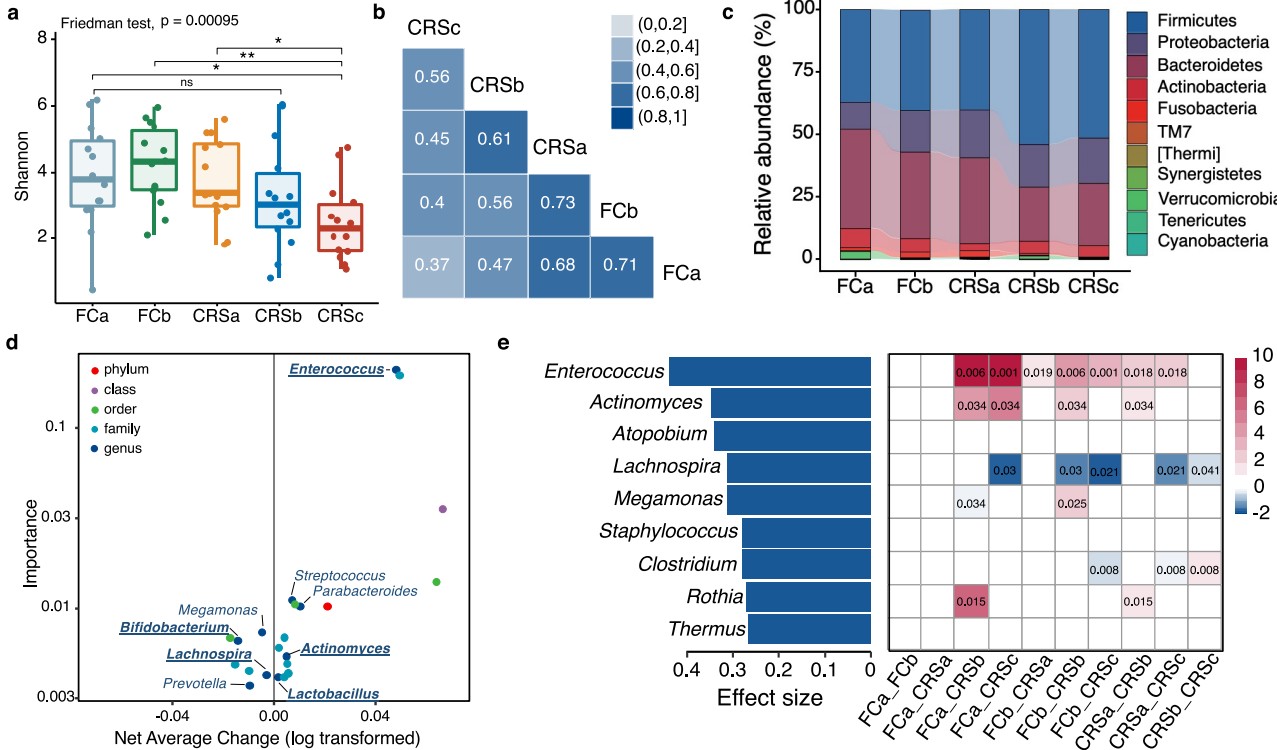

**Fig. 2 | Changes of microbial composition during CAR-T therapy in MM patients. a** Shannon diversity indices of gut microbiome across CAR-T stages in all myeloma patients. Differential tests by Friedman's tests and two-tailed Wilcoxon rank-sum tests for 10 pairwise comparisons of the five timepoints (*n* = 14). Bonferroni correction was applied for multiple testing; *FDR < 0.05, **FDR < 0.01. For FCa versus CRSc, adjusted *p* = 0.023; FCb versus CRSc, adjusted *p* = 0.009; CRSa versus CRSc, adjusted *p* = 0.017. Boxplots indicate the median (thick bar), first and third quartiles (lower and upper bounds of the box, respectively), lowest and highest data value within 1.5 times the interquartile range (lower and upper bounds of the whisker). **b** Pairwise Spearman correlation of OTU-level bacterial abundance across different timepoints. Rho value for each significant correlation is labeled inside box. **c** Stacked bar plot of mean phylum-level phylogenetic composition of bacterial taxa in myeloma patients across different therapy stages. **d** Significant

features identified by longitudinal analysis in Qiime2 "feature-volatility" plugin to identify taxonomic features associated with therapy stages. Scatter plot shows importance and average change of each important features by the longitudinal analysis. Genus-level features are labeled in the figure. Genus identified by both longitudinal analysis in Qiime2 and maSigPro are bolded and underlined. **e** Bar plot in the left shows significantly changed genera across the therapy identified by Friedman's tests (FDR < 0.05, *n* = 14). Effect size was estimated by Kendall's W Test. Heatmap in the right side denotes difference of each genus between two therapy stages. Red represents significant enrichment while blue represents significant depletion of the genus in the posterior stage comprising with the anterior stage. Significant *p* values were labeled in the boxes. Significances by two-tailed Wilcoxon rank-sum tests with FDR correction.

of OTUs, which belong to the phyla Firmicutes and Bacteroidetes and the orders Clostridiales and Bacteroidales. Cluster 2 was comprised of OTUs from a broader taxonomy, which included the orders Clostridiales, Bacteroidales, Lactobacillales, and Actinomycetales (Fig. 3f).

In genus level, we identified 30 genera with differential time-course patterns in MM patients with CR and PR (Fig. 4a left panel, Supplementary Data 6). To explore these differences further, we divided the therapeutic period into before and after CAR-T infusion and performed genus-level class comparisons using linear discriminant analysis (LDA) of effect size (LEfSe)[25] and generalized linear-mixed model (Fig. 4a middle and right panel). Consistent with the results from OTU-level pattern analysis, most of the significant genera such as *Faecalibacterium*, *Roseburia*, and *Ruminococcus* were enriched in CR patients after CAR-T. The genera *Bifidobacterium*, *Prevotella*, *Sutterella*, *Oscillospira*, *Paraprevotella*, and *Collinsella* had a higher abundance in CR versus PR patients both before and after CAR-T (Fig. 4a and Supplementary Fig. 5a). We also took patients with VGPR into consideration and analyzed the above-mentioned genera before and after CAR-T infusion. The bacterial abundance in VGPR patients fell somewhere between CR and PR patients, but no statistical significance was evident for most of genera (Fig. 4b and Supplementary Fig. 5b).

To explore whether early bacterial abundance was indicative of therapeutic response, we used RF feature selection to identify key discriminatory genera for responses[26]. By defining the stages before CAR-T

infusion as early, we applied feature selection procedures individually at both baseline (FCa) and post-chemotherapy (FCb) and identified gut microbiome signatures comprising 8 and 14 discriminatory genera separately for baseline and post-chemotherapy (Fig. 4c, d and Supplementary Fig. 5c). The area under the receiver operating characteristic curve (ROC) of the two RF models using these discriminatory features was 0.73 and 0.85, respectively (Fig. 4e, f). *Prevotella*, *Collinsella*, *Bifidobacterium*, and *Sutterella* were enriched in CR versus PR both before and after CAR-T infusion and were identified by RF analysis as significant at baseline and post-chemotherapy. This indicates potential associations between these genera and the response to CAR-T.

We also checked the abundance of these genera in r/r NHL and ALL patients. In NHL, *Faecalibacterium*, *Bifidobacterium*, and *Ruminococcus* were significantly (or almost significantly) enriched in CR versus PR and in patients not having a remission (NR), consistent with our results in myeloma (Supplementary Fig. 5e). However, for ALL, we observed enrichment of *Bifidobacterium, Roseburia*, and *Collinsella* in NR (Supplementary Fig. 5f), which differed from the results for MM and NHL but might be determined by the small NR sample.

In the independent 38 validation MM patients, no significance of Shannon diversity was observed between CR and PR (Supplementary Fig. 5g). Given that genus *Sutterella, Prevotella, Collinsella*, and *Bifidobacterium* were detected to be significant by both differential analysis and RF analysis at baseline and post-chemotherapy, we then examined

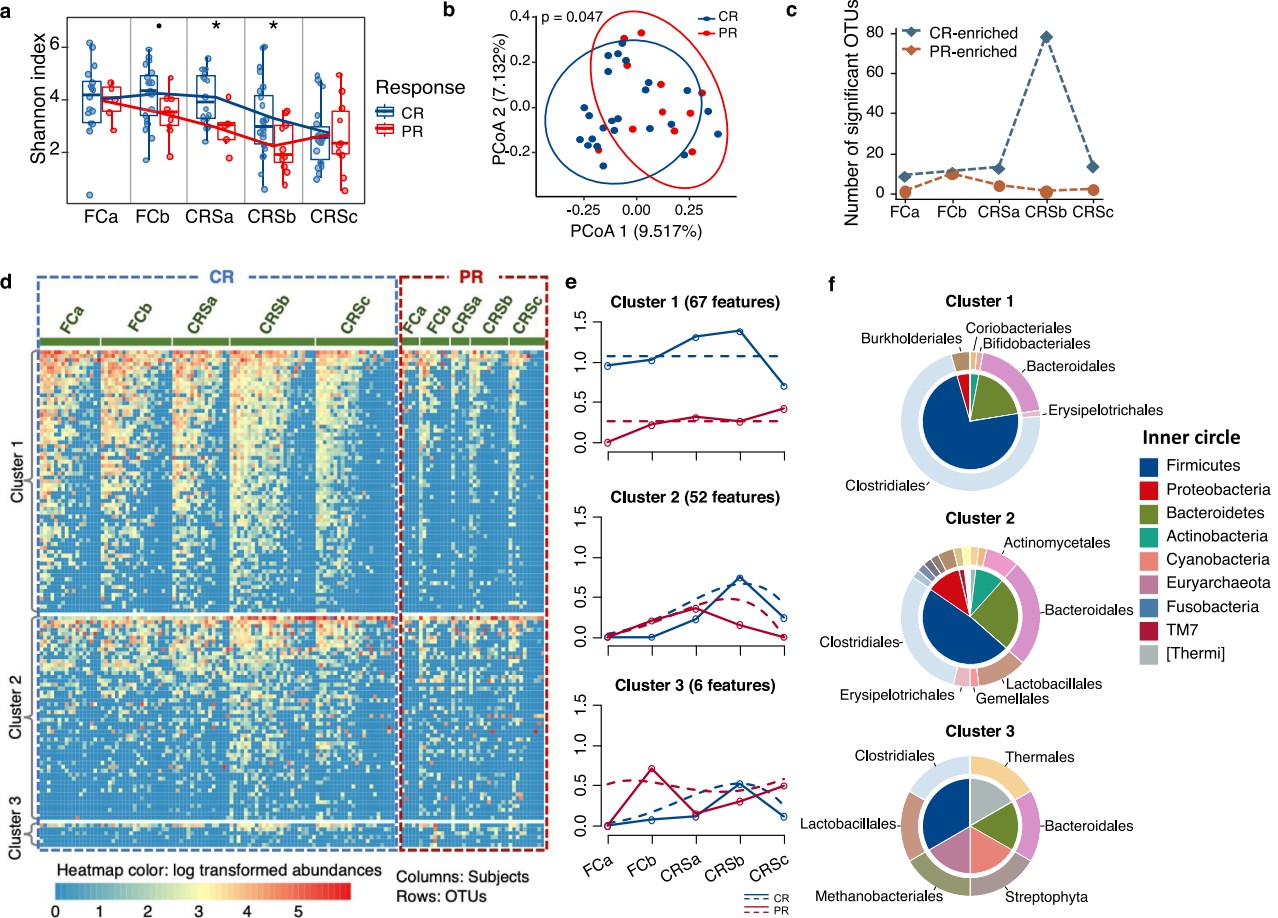

**Fig. 3 | Association of compositional differences in gut microbiome with responses to CAR-T therapy in MM patients. a** Shannon diversity indices of gut microbiome differed between CR and PR groups across CAR-T stages. Significances were assessed by two-sided Wilcoxon rank-sum test ($n = 35$). $P$ values were 0.077, 0.040, 0.036 for FCb, CRSa, and CRSb, respectively. Boxplots indicate the median (thick bar), first and third quartiles (lower and upper bounds of the box, respectively), lowest and highest data value within 1.5 times the interquartile range (lower and upper bounds of the whisker). **b** Principal coordinate analysis of fecal samples in CRSb stage by response (CR versus PR) using Canberra distance. $P$ value was calculated by PERMANOVA ($n = 35$). **c** Summary of number of PR or CR-enriched OTUs in different therapy stages. Difference between CR and PR groups was assessed by two-sided Wilcoxon rank-sum test. $P$ value significant cutoff was 0.05

($n = 35$). **d** Heatmap for abundance of OTUs with significant temporal differences between CR and PR groups identified by maSigPro (FDR < 0.05). Rows denote bacterial OTUs grouped into three sets according to regression coefficients and sorted by mean abundance within each set. Individual fecal samples were organized in columns and grouped by therapy stages. Columns in the blue and red dashed box show abundance and longitudinal changes of these OTUs in CR and PR groups across the five timepoints. Color of the heatmap is proportional to OTU abundance (red indicates higher abundance and blue indicates lower abundance). **e** Profiles of significant gene clusters correspond to **d**. Solid lines denote median profile of abundance of OTUs within cluster for each experimental group through time. Fitted curve of each group is displayed as dotted line. **f** Phylogenetic composition of OTUs within each cluster in **d** at phylum and order levels.

abundance of these significantly changed bacteria of interest in an independent 38 MM validation sample. We found that abundance of genera *Sutterella* and *Prevotella* were higher in CR group than that in non-CR group at multiple stages. No significance was observed for *Collinsella* and *Bifidobacterium* (Supplementary Fig. 5d).

To further demonstrate the association between these taxa and outcome, we assessed PFS following CAR-T therapy. By stratifying patients by tertile of bacterial abundance, we observed that for *Sutterella*, patients in the highest-abundance tertile had significantly prolonged PFS (Fig. 4g). Even after stratification by timepoints, this association remained significant (Supplementary Fig. 6a). However, for genus *Faecalibacterium*, which was reported to be significantly associated with PFS and anti-PD-1 therapy[19], we did not observe an association (Supplementary Fig. 6b, c).

### Associations between gut microbiome and CRS

Manifestations of severe CRS, namely high fever and greater amounts of cytokines, typically develop within several days after CAR-T cell

infusion and may cause death if untreated[27]. We scaled CRS from level 1 to 5[28]. To analyze associations between bacterial communities associated with CRS, we compared patients with severe (level 3) versus mild (level 1) CRS and severe and moderate CRS (level 2) in MM patients. We found 146 OTUs with different time patterns in the severe and mild groups (Supplementary Fig. 7 and Supplementary Data 7), and 99 OTUs with different patterns in the severe and moderate CRS groups (Supplementary Fig. 8 and Supplementary Data 8). The profiles of the OTU clusters for the comparisons were similar, with OTUs in clusters 1 and 3 having a higher abundance during late therapy in patients with severe versus mild CRS (Supplementary Figs. 7b and 8b).

By analyzing associations between CRS grade and taxa at the genus level, we identified signatures discriminating severe from mild CRS, including decreases in amount of *Bifidobacterium* and *Leuconostoc* in patients with severe CRS (Fig. 5a and Supplementary Data 9). *Bifidobacterium* was increased in patients with worse CRS, not only during the window of CRS, but also at early stages (Fig. 5a, b). *Leuconostoc* was significantly enriched during the window in patients with

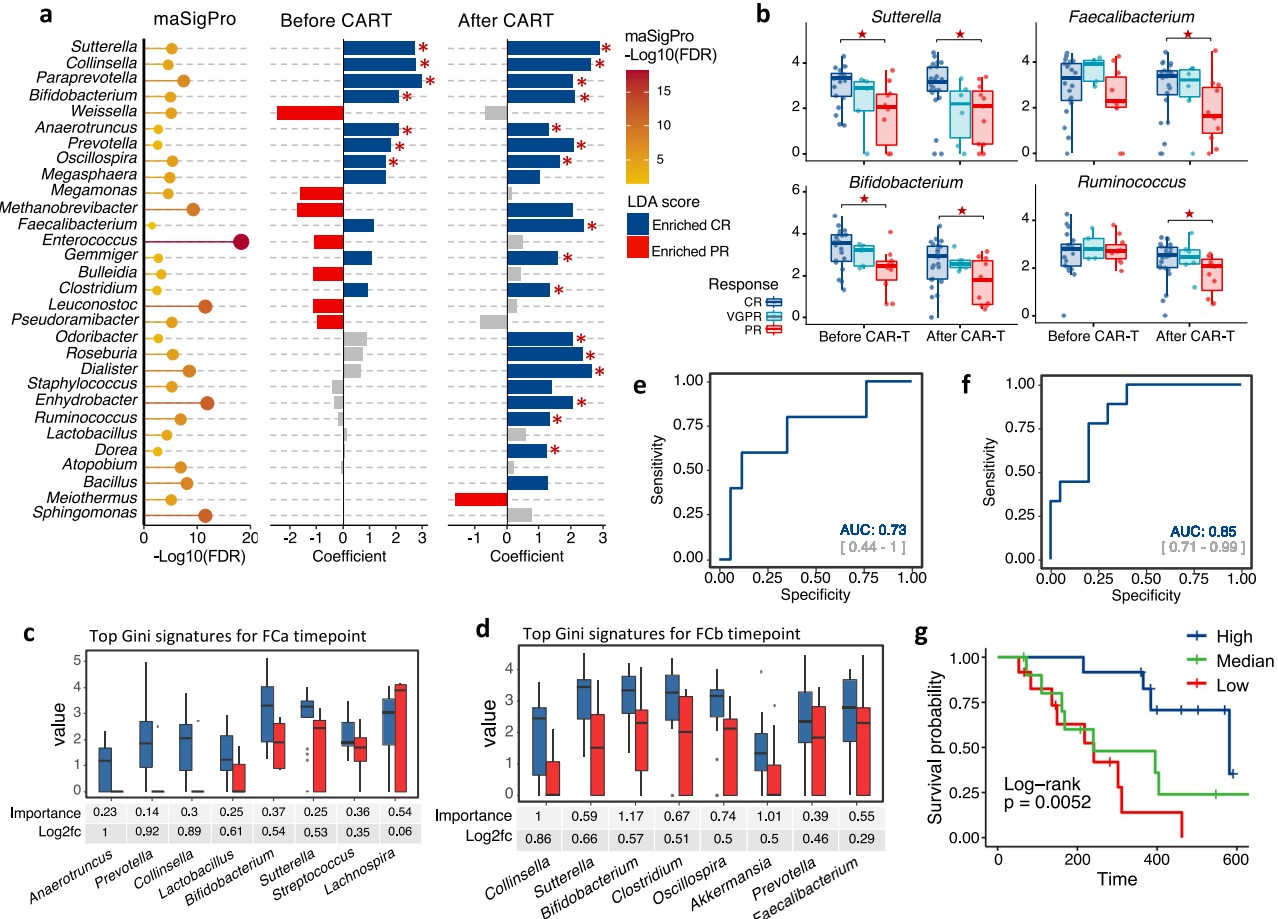

**Fig. 4 | Determination of correlated genera with clinical response to CAR-T therapy in MM patients. a** Differentially abundant genera between CR and PR group. Bubble plot in the left represents p values by maSigPro. Bar plots in the middle and right show significances and coefficients by generalized linear-mixed models (GLMMs) before and after CAR-T infusion (*n* = 35). Blue bars indicate significant enrichment in CR group while red bars indicate significant enrichment in PR group (FDR < 0.05). Red stars marked genera that was identified to be differentially abundant by linear discriminant analysis (*p* < 0.05 for Kruskal–Wallis H statistic and LDA score >2). *P* values by linear discriminant analysis for Sutterella, Collinsella, Paraprevotella, Bifidobacterium, Anaerotruncus, Prevotella, and Oscillospira before CAR-T were 0.0017, 0.0014, 0.038, 0.0015, 0.0064, 0.030, and 0.006, respectively; *P* values by linear discriminant analysis for Sutterella, Collinsella, Paraprevotella, Bifidobacterium, Anaerotruncus, Prevotella, Oscillospira, Faecalibacterium, Gemmiger, Clostridium, Odoribacter, Roseburia, Dialister, Enhydrobacter, Ruminococcus, and Dorea after CAR-T were 0.00012, 0.00076, 0.0060, 0.0.0067, 0.042, 0.0049, 0.011, 0.00017, 0.0035, 0.0058, 0.0073, 0.0013, 0.000038, 0.021, 0.0056, and 0.017, respectively. **b** Mean bacterial abundance [log2 (percentage + 1)] of CR, VGPR, and PR myeloma patents before and after CAR-T cell infusion (*n* = 43). Red stars indicate significant difference between CR and PR group by all three methods in panel **a**. *P* values for Sutterella by maSigPro were 1.17e-06, by generalized linear-mixed model were 7.86e-12 and 1.51e-14 before and after CAR-T, by linear discriminant analysis were 0.0017 and 0.00012 after CAR-T, respectively; *P* values for Faecalibacterium by maSigPro were 0.0093,

by generalized linear-mixed model and linear discriminant analysis were 1.22e-10 and 0.00017 after CAR-T, respectively; *P* values for Bifidobacterium by maSigPro were 2.19e-06, by generalized linear-mixed model were 5.67e-08 and 1.51e-08 before and after CAR-T, by linear discriminant analysis were 0.0015 and 0.0067 before and after CAR-T, respectively; *P* values for Ruminococcus by maSigPro were 1.49e-08, by generalized linear-mixed model and linear discriminant analysis were 0.00031 and 0.0056 after CAR-T, respectively. **c** Relative abundance [log2 (percentage + 1)] of top discriminative signatures at baseline (FCa) timepoint identified by RF feature selection procedure (*n* = 35). Genera with highest scores of mean decreases in Gini were selected. Importance scores in RF classification model and fold-change levels in log₂ scale are noted below plot for each genus. Blue and red colors indicate CR and PR group, respectively. **d** Same as panel **c** for post-chemotherapy (FCb) timepoint (*n* = 35). Only signatures enriched in CR patents are displayed. Those depleted in CR patents are displayed in Fig. S2C. **e** Receiver operating characteristic (ROC) curve of RF model using discriminatory genera as predictors for baseline timepoint. **f** Same as panel **e** for post-chemotherapy timepoint. **g** Kaplan–Meier (KM) plot of PFS curves by log-rank test for patients with high (dark blue), median (green), or low (red) abundance of *Sutterella*. Abundance of genus *Sutterella* was in terms of median abundance of all timepoints. Boxplots indicate the median (thick bar), first and third quartiles (lower and upper bounds of the box, respectively), lowest and highest data value within 1.5 times the interquartile range (lower and upper bounds of the whisker).

high CRS grade (Fig. 5a, b). In the 38 validation MM patients, no significance was observed for *Bifidobacterium* or *Leuconostoc* among different CRS grade groups (Supplementary Fig. 9).

## Correlation of gut microbial functions with CAR-T therapy
To determine if gut microbial functions correlated with CAR-T therapy, we first inferred community function of MM patients using Phylogenetic Investigation of Communities by Reconstruction of Unobserved

State (PICRUSt2). By applying time-course differential analysis, we identified differential pathways related to fatty acid metabolism, glutathione metabolism, quinone biosynthesis and glycan degradation (Supplementary Fig. 10) in the MM cohort. Further, we compared pathways across different CRS groups. Microbial function of fecal samples from patients with severe CRS had high metabolism or biosynthesis related to inflammatory compounds, including several pathways associated with phosphonate and its metabolism,

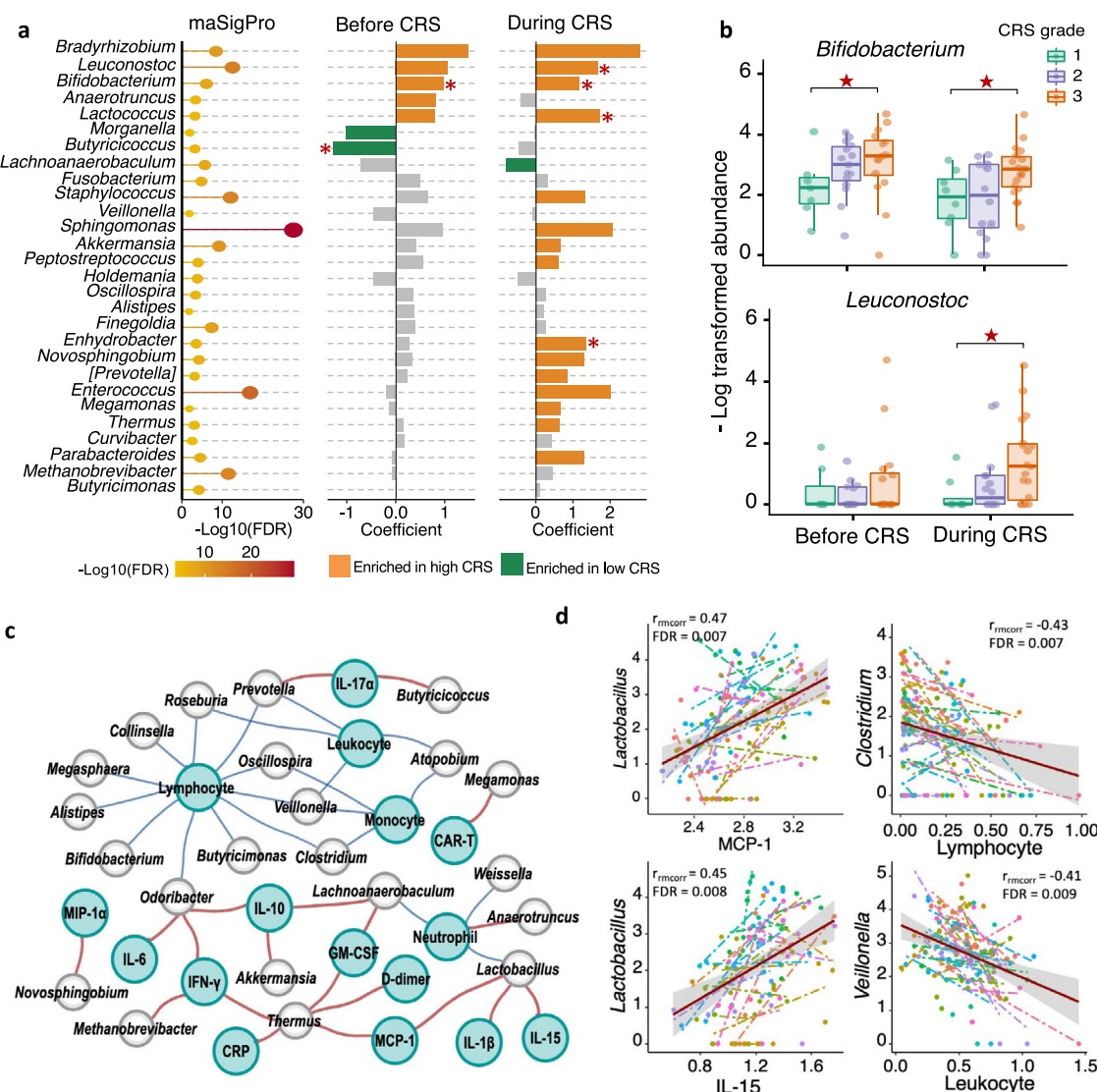

**Fig. 5 | Compositional differences between subjects with different CRS grades in MM patients. a** Correlation of differentially abundant genera with CRSgrade. Bubble plot in the left shows significant genera between severe and mild CRS groups by maSigPro (*n* = 27). Bar plots in the middle and right show significances and coefficients by generalized linear-mixed models (GLMMs) before and during CRS. Orange bars indicate positive correlation with CRS. Green bars indicate negative correlation. Red stars marked genera that was identified to be differentially abundant by linear discriminant analysis (*p* < 0.05 for Kruskal-Wallis H statistic and LDA score >2). *P* values by linear discriminant analysis for Bifidobacterium and Butyricicoccus before CAR-T were 0.003 and 0.027, respectively; *P* values by linear discriminant analysis for Leuconostoc, Bifidobacterium, Lactococcus, and Enhydrobacter after CAR-T were 0.016, 0.029, 0.0029, and 0.037, respectively. **b** Mean bacterial abundance in MM patients with different CRS grades before and during occurrence of CRS (*n* = 43). Red stars indicate significant difference between Grade 1 CRS and Grade 3 CRS group by all three methods in panel **a**. *P* values for Bifidobacterium by maSigPro was 8.9e-08, by generalized linear-mixed model were 9.75e-

06 and 1.42e-08 before and after CAR-T, by linear discriminant analysis were 0.003 and 0.029 before and after CAR-T, respectively; *P* values for Leuconostoc by maSigPro was 1.29e-14, by generalized linear-mixed model and linear discriminant analysis were 3.14e-11 and 0.016 after CAR-T, respectively. Boxplots indicate the median (thick bar), first and third quartiles (lower and upper bounds of the box, respectively), lowest and highest data value within 1.5 times the interquartile range (lower and upper bounds of the whisker). **c** Network representing correlations between gut microbes (gray nodes), immune cells and inflammatory markers (green nodes) at FDR < 0.05. Correlations were measured by repeated measure correlation analysis (rmcorr). Red edges indicate positive correlations and blue edges negative correlations. Edge width is proportional to correlation coefficient (*ρ*) calculated by Spearman correlation test. Only genera identified as associated with clinical response and CRS grade were included in correlation analysis. **d** Top 2 positive and negative correlations in repeated measure correlation analysis. Data are presented as mean ± SEM.

amino acid metabolism, lipoic acid metabolism, amino sugar, and nucleotide sugar metabolism and antibiotic synthesis (Supplementary Fig. 11).

Likewise, we performed differential analysis of PICRUSt2 predicted functions in the 38 validation MM cohort. Comparing PR with CR, differential pathways concerning glutamate (ᴅ-Glutamine and ᴅ-glutamate metabolism), glycan (Glycan biosynthesis and metabolism), arginine, proline (ᴅ-Arginine and ᴅ-ornithine metabolism, Arginine and proline metabolism) and phenylalanine (phenylpropanoid

biosynthesis) were revealed (Supplementary Fig. 12a), among which the pathways related to glutamate and phenylalanine metabolism were endorsed in differential analysis of predicted KEGG pathways between PR and CR groups in the discovery MM sample (Supplementary Fig. 10a). Lipopolysaccharide and steroid biosynthesis pathways were also consistently found to be differ between the CR and PR group by differential analysis of predicted pathways (Supplementary Fig. 10a) and metabolites (Supplementary Fig. 12a). Referring to the CRS grade-related pathway, difference in glycerolipid metabolism pathway was

reproducible detected in both the discovery (Supplementary Fig. 11a) and validation MM samples (Supplementary Fig. 12c).

In addition, we applied metabolic Liquid Chromatography Mass Spectrometry (LC-MS) to quantify concentration of fecal metabolites during CRS. Intermediates (Choline, L-Cysteine, S-Sulfo-L-cysteine, Rosmarinic acid, L-Phenylalanine, and 2-Phenylacetamide) involved in multiple amino acid metabolism pathways were differentially abundant between CP and PR group when during CRS (p-value < 0.05). We also identified metabolites concerning phosphonate and phosphonate metabolism (Bialaphos) and steroid biosynthesis (Desoxycortone) to be differ between CR and PR (Supplementary Fig. 13). In differential analysis between CRS groups, we identified phosphocreatine which annotated to arginine and proline metabolism (Supplementary Fig. 14). Moreover, three abovementioned pathways (i.e., tyrosine metabolism, phenylalanine metabolism, phosphonate, and phosphonate metabolism) were also indicated to have differentially abundances between the CR and PR group in the predicted pathway analysis (Supplementary Fig. 10). Two pathways (tyrosine metabolism and phenylalanine metabolism) were also differed among patients with different CRS grades (Supplementary Fig. 11). Additionally, we performed pathway enrichment analysis of differentially abundant metabolites between the CR and PR subjects to reveal distinction on metabolic functions (Supplementary Fig. 15a). Two pathways (Phenylalanine, tyrosine and tryptophan biosynthesis; Riboflavin metabolism) reached marginal significance (p = 0.07). These concordant findings strengthened the results of functional prediction analysis and highlighted the importance of amino acid metabolism during the CAR-T therapy.

### Correlation of CRS-related cytokines with gut microbes

Primary inflammatory markers of CRS are cytokines, such as IL-6, IL-2, IL-10, interferon gamma (IFN-γ), and tumor necrosis factor-α (TNF-α). Various cytokines are elevated in the serum of patients experiencing CRS after CAR-T cell infusion[29]. By assessing serum cytokine concentrations and immune cell numbers during CAR-T, we observed significantly increased amounts of serum inflammatory cytokines (IL-6, CRP, IFN-γ, D-dimer, ferritin) but low numbers of immune cells (monocytes, lymphocytes, neutrophils, leukocytes) in severe CRS (Fig. 5c). We also compared serum cytokine concentrations and immune cell numbers in CR and PR, observing significant differences for many of them (see Supplementary Fig. 16).

To explore further associations between the gut microbiome and CRS during CAR-T therapy, we determined whether serum cytokine concentrations and numbers of PB immune cells correlated with the abundance of gut microorganisms (Fig. 5d). By assessing common within-individual correlation for repeated measures[30], we constructed correlation network between gut microbes, cytokines, and immune cells (Fig. 5c). The top significant correlation pairs were MCP-1 and *Lactobacillus*, lymphocyte and *Clostridium*, IL-15 and *Lactobacillus*, leukocyte and *Veillonella* (Fig. 5d). In addition, serum level of lymphocyte was negatively correlated with 11 genera, including multiple genera related to CRS level such as *Bifidobacterium, Butyricimona and Oscillospira*. M1 and M2 macrophages, which play a key role in CRS initiation, did not show significant correlation with any microbes.

## Discussion

Although several studies have revealed the critical role of the gut microbiome in treatment responses and survival after administration of another important immunotherapy — immune checkpoint inhibitor (e.g., PD-1, PD-L1) therapy[20], no study has reported on the association between the gut microbiome and CAR-T therapy. In this study, we describe the changes of the gut microbiome during CAR-T therapy and associations with treatment responses and CRS severity in CAR-T-treated patients with B-cell malignancies. Although neurotoxicity is another major toxicity associated with CAR-T cell therapy, we were not able to analyze the microbiome in relation to neurotoxicity incidence

and severity because of very limited patients presented in the MM patients. Future work with larger number of patients is needed in order to explore the relationship between microbiome and neurotoxicity after CAR-T cell treatment.

Some of the bacterial genera with differences in abundance in CR versus PR patients have been reported to be involved in the regulation of the immune response, including to immunotherapy. *Faecalibacterium*, reported to enhance antitumor immune responses and survival after anti-PD-1 therapy in melanoma[19, 31], was in this study associated with CR. Multiple species within the genera *Bifidobacterium* and *Collinsella* increased in responders to anti-PD-1 therapy for melanoma[32], resulting in depleted peripherally derived colonic regulatory T cells, increased Batf3-lineage dendritic cells (DC), and augmented T-helper 1 cell (Th1) responses and thus better immune-mediated tumor control[33]. Here, we observed an increased abundance of these two bacteria in CR patients, suggesting a similar response-associated effect of these taxa on the immune system across cancer types and therapeutic strategies. Moreover, in a recent study, taxonomic analysis by 16 S sequencing of the fecal microbiome showed that members of *Ruminococcus*, *Faecalibacterium*, and *Bacteroides* were associated with response to CD19 CAR-T cell therapy. Higher abundances of microbial taxa with the class Clostridia, including the genera *Ruminococcus*, *Faecalibacterium* were associated with day 100 complete response and no toxicities in patients with ALL and NHL after receiving CD19 CAR-T cell treatment were observed[34].

Nevertheless, some taxa might have effects that are specific for cancer or therapy types. For example, high abundance of genus *Sutterella* was associated with both CR and prolonged survival after CAR-T therapy. However, previous studies reported higher numbers of *Sutterella* in non-responders versus responders in non-small-cell lung cancer (NSCLC) treated with nivolumab[35]. Besides, in this study, we observed contradictory results for the genus *Bifidobacterium*, *Roseburia*, and *Collinsella* in three types of hematologic malignancy (Supplementary Fig. 2f). This indicates a potentially distinct involvement or function of some bacteria in different cancer types and treatments. But these findings require confirmation in studies with larger cohorts.

Gut microbial communities contribute to inter-individual variation in cytokine responses[36]. In this study, *Bifidobacterium* and *Leuconostoc* were enriched in myeloma patients with severe CRS while *Butyricicoccus* were enriched in patients with mild CRS. *Bifidobacterium* is a commonly used probiotics to enhance host immune. Another report demonstrated that *Bifidobacterium* correlated with the production of multiple cytokines (e.g., IFN-γ) in a stimulus-specific pattern[36]. Other oral supplementation tests concluded that *Bifidobacterium* could enhance nonspecific cellular immune response by activation of immune cells and release of various cytokines[37]. Moreover, it is worth noting that species composition of the *Bifidobacterium* flora will affect cytokine production. Cultivation of murine macrophage-like cells with divergent *Bifidobacterium* strains differentially stimulated production of proinflammatory TNF-α, IL-1β, and IL-6[38]. Thus, we proposed that the increased *Bifidobacterium* abundance in patients with server CRS may compose of more diverse *Bifidobacterium* strains predisposing toward CRS in CAR-T therapy. Further studies should pay attention to species and strain level composition of *Bifidobacterium* and their association with CRS. Besides, all patients with severe CRS in this study were treated with tocilizumab and/or corticosteroid. In future work, we will apply the CRS mice model to investigate the effects of Tocilizumab and corticosteroid on gut microbiome after CAR-T cell treatment.

The mechanisms through which gut microbes modulate host immunity are largely unknown. Gut microbial communities modulate host defenses mainly through the release of intermediary metabolites rather than by direct interaction between specific microorganisms and immune cells[36]. Multiple bioactive gastrointestinal metabolites produced by gut microbes, such as amino acids, short-chain fatty acids

(SCFAs; e.g., butyrate), and bile acids, exert immunomodulatory functions through immune cell metabolic reprogramming or transcriptional and epigenetic modulation of immune-related genes[39]. Lipopolysaccharide (LPS) from some pathogens is a well-known endotoxin that can stimulate the release of a variety of cytokines/chemokines[40, 41]. Peptidoglycans in bacterial cell walls are a conserved PAMP that trigger innate inflammatory responses throughout the body[42].

By comparing community functions inferred from PICRUSt2, we found pathways concerning amino acid metabolisms to have differentially abundances in samples with distinct treatment outcomes and CRS grades in myeloma patients. Phenylalanine metabolism pathways were suggested to be differentially abundant between CR and PR patients in both discovery and validation MM patients through differential analysis of predicted functions and metabolites. Differential analysis of LC-MS-based fecal metabolites revealed that intermediate metabolites of phenylalanine metabolism, L-Phenylalanine and 2-Phenylacetamide, were all enriched in the CR versus PR patients. The roles and underlying mechanisms of intestinal phenylalanine in regulating immunotherapy worth further investigating. Amino acids, such as cysteine, glutamine, phenylalanine, tryptophan, and arginine, play an important role in regulating immune responses by activating T lymphocytes, B lymphocytes, natural killer cells and macrophages, regulating cellular redox state, gene expression and lymphocyte proliferation, and stimulating production of antibodies, cytokines, and other cytotoxic substances[43, 44]. Gut microbiota is an essential mediator that keeps homeostasis of host amino acids. In the intestine, gut microbes help to metabolize amino acids from food or synthesize several essential amino acids de novo and, in turn, regulate innate and adaptive immunity of hots[45]. Compositional difference of gut microbiome may thus lead to disparate clinical outcomes and CRS for the CAR-T therapy. However, arginine and tryptophan metabolism, which correlated with CRS in COVID-19[46], were not identified in this study. This might be due to technical limitation of amplicom sequencing and function inference of PICRUSt2. Metagenomic sequencing is thus recommended to further explore association of bacterial functions with CAR-T therapy.

In this report, we observed significant correlation of gut microbiome with treatment outcome and CRS grade of CAR-T therapy. To further demonstrate the correlation and investigate underlining mechanisms, more wet-lab experiments are needed in future. Germ-free mice are helpful tools to validate function of gut microbiome. For example, fecal microbiota transplantation (FMT) from CR and PR patients into germ-free mice to construct mouse colonized by donor microbiota. Then physiological and biochemical response of these FTM-treated mice to myeloma cells and CAR-T cells could be surveyed. Moreover, gut microbiota could be damaged by antibiotic treatment and rescued by FMT, which could help to validate the function of gut microbiome in immunotherapy. On the other hand, genus Sutteralla was found to be important biomarkers for treatment outcome. Further study should demonstrate the predictive role of Sutteralla in a larger cohort of myeloma and explore its predictive capacity in CAR-T therapy of other types of tumors. To study correlation of genus Bifidobacterium with CRS grade, probiotic supplement of Bifidobacterium species to myeloma or CAR-T mouse model could help to reveal immune responses caused by Bifidobacterium. Other oral supplement strategies include bacterial metabolites, such as amino acids, fatty acids, cytotoxin, could also be used to demonstrate mechanism underlying the effect of gut microbes on the CAR-T therapy.

In addition to myeloma, CAR-T therapy has been applied to other blood cancers and solid tumors. The link between the gut microbiome and different cancer types needs to be studied systematically. Our research describes associations between changes in the gut microbiome of CAR-T patients and clinical responses and survival. This will open an avenue for investigating the interaction of the gut microbiome and CAR-T cells and lead to novel ways to improve the therapeutic efficacy of CAR-T therapy by targeting the gut microbiome.

As one of the most prominent treatment strategies for hematologic malignancies, CAR-T cell therapy has recently received great attention. Here for the first time, we found that the dynamic changes in the gut microbiome correlated significantly with therapeutic response and CRS during CAR-T treatment of hematologic malignancies (B-ALL, B-NHL, and MM). These findings will aid the development of novel biomarkers for predicting treatment outcome and CRS severity, thereby optimizing the management of these patients while reducing potential toxicities.

## Methods
### Study design and protocol
The study was approved by the Institutional Review Board of the First Affiliated Hospital, School of Medicine, Zhejiang University and was registered in the Chinese Clinical Trial Registry (ChiCTR1800017404). All patients provided written informed consent for participation in accordance with the guidelines of the Declaration of Helsinki and signed agreement for collection and analysis of microbiome samples. Patient inclusion criterias were: (1) age <75 years; (2) relapsed or refractory BCMA−positive MM before CAR-T cell treatment; and (3) expected survival > 12 weeks and adequate performance status and organ function to tolerate treatment. Exclusion criteria were: (1) pregnancy or lactation; (2) having received systemic (except inhaled) steroids in the previous 2 weeks or gene therapies; (3) having medical conditions such as severe mental illness, clinically significant cardiovascular disease, severe renal or hepatic dysfunction, or active infection; and (4) any conditions that might increase treatment risks. Data and sample collection were carried from 1 July 2018 to 30 September 2021. Patient information and the methods related to two types of cancer (ALL and NHL) are presented in the Supplementary Materials.

Peripheral blood mononuclear cells (PBMCs) were obtained from each patient by leukapheresis for CAR-T cell preparation. The purified CD3+ T cells were transduced with lentiviral vector to express BCMA CAR (Fig. 1b). Then the engineered T cells were expanded ex vivo under interleukin-2 stimulation. All patients received lymphodepletion with fludarabine (30 mg/m$^2$ of body surface area daily on days −4, −3, and −2) and cyclophosphamide (500 mg/m$^2$ daily on days −3 and −2) followed by an infusion of BCMA CAR-T cells on day 0[24]. The primary response outcome, defined by the guidelines from the International Myeloma Working Group (IMWG) as a complete response (CR), very good partial response (VGPR), or partial response (PR) in the third month after CAR-T treatment[47, 48]. CRS was graded by the Lee criteria[28].

### Microbiome sample collection and restoration
Gut microbiome samples were collected at five timepoints (Fig. 1c). All fecal samples were collected with the GUHE Flora Storage kit (Zhejiang Hangzhou Equipment Preparation 20190682, GUHE Laboratories, Hangzhou, China), which maintains microbial DNA stability at room temperature for as long as 1 month. All samples were frozen at −80 °C prior to DNA extraction. The stages of FCa, FCb, and CRSa were defined as early stages and CRSb and CRSc as late stages. The CRS grade 1 was defined as Mild, CRS grade ≤2 as Moderate, and CRS grade ≥3 as Severe.

### Assessment of serum cytokine concentrations
All blood samples were stored at 4 °C until centrifugation at 5000 rpm for 6 min. The supernatant liquids were quantified with the BD Cytometric Bead Array Human Th1/Th2/Th17 Cytokine Kit and its corresponding software (BD Biosciences) according to the manufacturer's instructions. Plasma levels of MIP-1α, GM-CSF, MCP-1, IL-15, IL-1β, IL-1α,

and IL-17α were determined by the Bio-Rad human Multi-cytokine detection array.

## Assessment of CAR-T cell expansion and persistence

Serial PB samples were collected in BD Vacutainer $K_2$EDTA tubes (BD Biosciences) before and after CAR-T cell infusion. The expansion of CAR-T cells in vivo was determined by detecting the CAR-T ratio continuously in PB as described[49, 50]. BCMA CAR-T expression was assessed using biotin-SP-conjugated F(ab')2 fragment goat anti-mouse IgG, F(ab')2 fragment-specific antibody, and the secondary staining reagent streptavidin-FITC (BioLegend, 405202) or streptavidin-PE (BioLegend,405204) using a dilution of 1:50. The flow cytometry gating strategy is presented in Supplementary Fig. 17.

## DNA extraction

Total bacterial genomic DNA samples were extracted using the MO BIO PowerSoil DNA Isolation Kit (MO BIO Laboratories, Carlsbad, CA, USA). The quantity and quality of extracted DNA was assessed using both the NanoDrop ND-1000 Spectrophotometer (Thermo Fisher Scientific, Waltham, MA, USA) and agarose gel electrophoresis.

## Bacterial 16S rRNA gene sequencing

The V4 region of the 16S rRNA gene was amplified with bacterial universal primers: 515 F (5'-GTGCCAGCMGCCGCGGTAA-3') and 806R (5'-GGACTACH VGGGTWTCTAAT-3'). The primers used for amplification contain adapters for the HiSeq platform and single-end barcodes allowing pooling and demultiplexing sequences of PCR products. Amplified sequences were purified with AMPure XP beads (Agencourt, Inc, Beverly, Manchester, MA, USA) and AxyPrep DNA Gel Extraction Kit (Axygen, Inc, Union City, CA). Qualified PCR products were sequenced with the HiSeq platform (Illumina, Inc, San Diego, CA, USA) using the 2 × 150-bp paired-end sequencing protocol.

## Amplicon data processing

Sequenced reads were demultiplexed according to barcodes. Paired-end reads were merged with the *fastq_mergepairs* command from VSEARCH (v. 2.4.4)[51].The minimum length of overlap between paired-end reads was set to 5. Merged reads were then imported into Qiime2 (v. 2020.2)[52]. Jointed reads were processed by the *qiime quality-filter q-score-joined* command to filter sequences with low-quality scores. Sequences were denoised with the *Deblur* workflow[53]. Amplicon sequence variants (ASVs) were summarized with the *feature-table summarize* command. To calculate phylogenetic diversity, a rooted phylogenetic tree was constructed using the *align-to-tree-mafft-fasttree* pipeline from the *q2-phylogeny* plugin within Qiime2. The pipeline performed a multiple sequence alignment of the ASV sequences and then masked the alignment to remove positions that are highly variable. The masked alignment was used to generate a phylogenetic tree by *FastTree* program[54]. Alpha and beta diversity matrices were generated through the *q2-diversity* plugin using the above-mentioned ASV feature table and rooted phylogenetic tree. De novo clustering of ASVs was performed with the *cluster-features-de-novo* command within *vsearch* plugin[51]. Input features were collapsed at 97% identity, resulting in new OTU features that are clusters of the ASV features. To annotate the OTUs, we downloaded the pre-trained Naive Bayes classifier trained on the Greengenes 13_8 99% OTU database, which was provided by developers of Qiime2 (https://docs.qiime2.org/2020.2/data-resources/). Representative OTU sequences were then annotated with pre-trained Naive Bayes classifier trained on the Greengenes 13_8 99% OTU database using the *feature-classifier* plugin[55]. The sequences used for training were trimmed to include only the V4 region. Taxonomic composition was summarized with the *collapse* method from the *taxa* plugin within Qiime2.

## Functional prediction

We used the OTU feature table generated from Qiime2 to predict microbial community function with PICRUSt2 (v.2.3.0-b)[56]. PICRUSt2 integrated more than 40,000 bacterial and archaeal genomes from the Integrated Microbial Genomes (IMG) database and pre-calculated gene contents for each organism to generate a table of predicted gene family abundances for each organism. Then functional prediction procedure was performed based on the precalculated gene content table and 16 S rRNA marker gene sequencing profile of each sample. The algorithm searched for the most closely related organisms with annotated genomes in the gene content table for each 16 S rRNA marker gene sequence to infer gene contents per sample. Gene family abundance per sample was summarized and grouped into KEGG orthologs (KOs). To facilitate the interpretation of functional results, KOs were further summarized into KEGG pathways on the basis of structured pathway mappings. For differential pathway analysis, we applied the two-sided Welch's *t*-test to identify discriminative KEGG pathways concerning clinical responses (PR versus CR) and CRS level (level 1 versus level 3).

## Bioinformatics and statistical analysis

Comparisons of alpha diversity and taxonomic abundances between two groups were conducted with the Wilcoxon rank-sum test, while comparisons among three or more groups were conducted using the Kruskal-Wallis rank-sum test. For beta diversity analysis, a PCoA plot was generated with weighted Unifrac distances. To test the significance of between-sample diversity alternation, permutational analysis of variance (PERMANOVA) was performed with the *adonis* function within the R (v. 3.6.2) package *vegan* (v.2.5-6). Clinical data were analyzed using *SPSS* software (v. 23.0). Flow cytometry data were analyzed using FlowJo 10 software.

The *feature-volatility* plugin[57] within Qiime2 was applied to implement longitudinal analysis to identify features that are associated with therapy stages. In this pipeline, supervised learning regressor was used to identify important features and assess their ability to predict therapy states. Unclassified taxonomic features, features absent in more than 90% of all samples, and features with low abundance (<0.01%) were all excluded from the analysis. Net average change scores and importance scores, which denote the correlation between input features and therapy stages, were exported and visualized in a volcano plot. Only features with net average change scores more than 0.2% and importance scores within the first tertile of distribution were retained. Considering repeated measurements, we additionally performed Friedman's test (Stats package v.3.6.2) with post-hoc multiple comparison testing of pairwise combinations for longitudinal analysis of diversity and bacterial taxa. For multiple testing correction, false discovery rate (FDR) was calculated (Stats package v.3.6.2).

For time-course differential analysis, the R package *maSigPro* (v.1.58.0)[58, 59] was used to find taxonomic features with significant temporal changes and significant differences between experimental groups (e.g., clinical response and CRS grade groups). Specifically, the *maSigPro* algorithm defined a generalized regressive model by dummy variables followed by two regression steps: the first one selects features with non-flat profiles by the least-squared technique and the second step creates best regression models for each feature by using stepwise regression to identify features with different profiles between experimental groups. We used as input, the normalized relative abundance (scaled to 100 million) and excluded features that did not occur in more than 90% of all samples. We employed a negative binominal regressive model for the microbial counts data and ran *maSigPro* on therapy stages with a degree of 4. All features with a significant group difference were exported. The significant features were further clustered together using the *hclust* function (Stats package v.3.6.2) method according to the patterns of their relative abundance. For each cluster, a median profile and fitted curve of all

included features were summarized to visualize the profile pattern. For genera that was identified by maSigPro, we applied generalized linear-mixed models (GLMMs, Stats package v.3.6.2)) to identify genera with different abundances before and after CAR-T infusion. For multiple testing correction, FDR was applied.

The Linear Discriminant Analysis (LAD) effect size (LEfSe, v.1.1) algorithm[25] was employed to identify differentially abundant features between groups (e.g., between clinical response and CRS grade). The method first detected features with significant differential abundance using the non-parametric factorial Kruskal–Wallis rank-sum test with pre-defined $\alpha$ of 0.05. Significant features were then used to build a LDA model for estimating the effect size of each differentially abundant feature. The LDA score threshold for discriminative features was set to 2.0.

To identify early predictive biomarkers with respect to clinical response (PR vs. CR), we implemented a random forest (RF) feature selection procedure within the R package *caret* (v. 6.0-85). The recursive feature elimination (RFE) algorithm in caret package with 5-fold cross validation was applied for feature selection. An optimized number of feature sets was determined by performance of 5-fold cross validation. To depict the receiver operating characteristic (ROC) curve and calculate the area under the curve (AUC), the *pROC* package (v.1.16.1) was utilized.

For PFS analysis, subjects were classified as high, medium, or low based on tertiles of the distribution of specific taxa abundance (e.g., genus *Sutterella*). Time to progression was defined as the interval (in days) from the date of CAR T-cell infusion to the date of disease progression. Survival curves were estimated using the Kaplan–Meier product-limit method and compared using the log-rank test within the R package *survminer* (v.0.4.7).

We applied repeated measures correlation (rmcorr, v.0.4.5) analysis to test the association between bacterial abundance and concentration of immune cells and inflammatory factors. Only genus-level features deemed to be associated with clinical response and CRS grades were included in this analysis. Associations with FDR less than 0.05 were depicted using Cytoscape (v.3.9.0)[60].

## Reporting summary

Further information on research design is available in the Nature Research Reporting Summary linked to this article.

## Data availability

The raw sequence reads used in this study have been deposited in the Sequence Read Archive (SRA) of the NCBI under accession number PRJNA813944 All software packages used for the study are publicly available. Source data are provided with this paper.

## Code availability

Scripts used to produce figures, alongside scripts used in data analysis are available at: https://github.com/jjlea/CART-microbiome.

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

## Acknowledgements

This study was supported by the National Natural Science Foundation of China (grant no. 81730008 and 81770201), Key Project of Science and Technology Department of Zhejiang Province (grant no. 2019C03016), China Precision Medicine Initiative (2016YFC0906300), Research Center for Air Pollution and Health of Zhejiang University, and the State Key Laboratory for Diagnosis and Treatment of Infectious Diseases, The First Affiliated Hospital of Zhejiang University. We thank Dr. David Bronson and Mrs. Judith Gunn Bronson of Stem Line Publishing Services, Inc. for their editing of this manuscript.

## Author contributions

H.H. designed and supervised the clinical study; M.D.L. designed and supervised data analysis; M.M. designed the clinical study. H.C., Y.Z., and Y.W. supervised the CAR T-cell production; Y.H., W.W., M.Z., G.W., R.H., H.Z., and L.W. collected clinical data; M.D.L., J.L., F.N., Z.Y., H.G., X.G., Z.B., and W.B. analyzed data, wrote, and revised the manuscript; J.L., Z.Y., and H.G. performed statistical analyses; Y.H., W.W., M.Z., G.W., R.H., and L.W. enrolled patients and took care of the patients; H.H., M.D.L., M.M., A.N., and D.B. revised the manuscript.

## Competing interests

The authors declare no competing interests.

## Additional information

[1]Bone Marrow Transplantation Center, The First Affiliated Hospital, School of Medicine, Zhejiang University, Hangzhou, China. [2]Zhejiang Province Engineering Laboratory for Stem Cell and Immunity Therapy, Hangzhou, China. [3]Institute of Hematology, Zhejiang University, Hangzhou, China. [4]Zhejiang Laboratory for Systems & Precision Medicine, Zhejiang University Medical Center, Hangzhou, China. [5]State Key Laboratory for Diagnosis and Treatment of Infectious Diseases, National Clinical Research Center for Infectious Diseases, Collaborative Innovation Center for Diagnosis and Treatment of Infectious Diseases, The First Affiliated Hospital, Zhejiang University School of Medicine, Hangzhou, China. [6]Research Center for Air Pollution and Health, Zhejiang University, Hangzhou, China. [7]Department of Hematology, Sorbonne University, Hospital Saint Antoine, Paris, France. [8]INSERM UMRs 938, and EBMT Paris Study office/CEREST-TC, Paris, France. [9]Hematology and Bone Marrow Transplantation Division, Chaim Sheba Medical Center, Tel-Hashomer, Israel. [10]Clinical Translational Research Center, Shanghai Pulmonary Hospital, Tongji University School of Medicine, Shanghai, China. [11]Department of Immunology, Sloan Kettering Institute, Memorial Sloan Kettering Cancer Center, New York, NY, USA. [12]These authors contributed equally: Yongxian Hu, Jingjing Li, Fang Ni, Zhongli Yang. ✉e-mail: alexhchang@yahoo.com; vandenbm@mskcc.org; ml2km@zju.edu.cn; huanghe@zju.edu.cn

