## [Peer Review File · Nature Communications]

CAR-T cell therapy-related cytokine release syndrome and therapeutic response is modulated by the gut microbiome in hematologic malignanciesREVIEWER COMMENTS

Reviewer #1 (Remarks to the Author):

The manuscript by Hu et al presents the results of longitudinal fecal 16S rRNA gene sequencing of MM (n=43), B-ALL (n=23), and B-NHL (n=12) subjects receiving CART therapy (5 time points). The authors observed changes in microbiome composition and reduced microbial diversity in MM patients after CART therapy, with similar trends in B-ALL and B-NHL. The authors also reported differences in microbial composition and predicted function across the time points between MM patients with complete response (CR) compared to those with partial response (PR). Microbes including *Sutterella* were reported to be associated with clinical response. Differences in microbial composition and predicted function were also reported between mild and severe cytokine release syndrome (CRS).

This study is motivated by prior research demonstrating an important role for the gut microbiome in modulating immunotherapy response. Extrapolating from this literature, it is probable that the microbiome can be affected by CART therapy and that the microbiome (either baseline profiles or alterations in response to therapy) could influence response to CART therapy. Hu et al use an appropriate strategy to address this important unanswered question: a longitudinal microbiome association study of CART recipients with clinical outcome monitoring. The findings of this manuscript support the concept that the microbiome could be linked to CART response, which would have important implications for the field. However, this study has critical limitations that reduce its impact:

- 1) Small study size: Most of the results presented in this manuscript are based on the 43 patients with MM. This is a modest size for a microbiome study and raises concerns about the robustness and reproducibility of the reported findings. In particular, the comparisons of complete responders vs. partial responders only utilize 24 and 11 subjects, respectively, and comparisons of grade 1 vs. grade 3 CRS utilized 8 and 19 subjects, respectively. With such small numbers, it is likely that the differential taxa and predictive models shown by the authors are overfitted and won't be reproducible.
- 2) Absence of a validation cohort: Given the great heterogeneity of findings across small microbiome studies, it's important for microbiome association studies to demonstrate the reproducibility of key findings in an independent validation cohort. This is particularly critical here given the limited number of subjects in the CR vs. PR and CRS grade comparisons. The B-ALL and B-NHL subsets could not be used to validate the CR vs. PR analyses as there were only 2 or 3 subjects with PR, respectively.
- 3) Statistical issues: The study utilizes a longitudinal sample collection strategy but none of the analyses account for the repeated measures design and thereby inflate significance due to positive correlation of the repeated measurements (examples provided under additional comments). Also, the manuscript does not give any indication that p-values were adjusted for multiple hypothesis testing, which is critical given the large number of microbial taxa and predicted pathways that were tested. In addition, none of the analyses were adjusted for covariates (e.g. age, sex, antibiotic use before or during treatment, prior autologous stem cell transplantation, etc.) which could confound the association of microbiota with outcomes.
- 4) 16s sequencing with imputed metagenomics was used rather than shotgun metagenomics sequencing: The study would have been strengthened by the use of shotgun metagenomics (as has been used in the prominent immunotherapy microbiome studies), which would allow improved species and possibly strain level taxonomic assignment as well as more accurate assessment of microbiome functional capacity.

Additional comments:

- It's not indicated how many samples were collected at each of the five time points from the various subgroups.
- What was the median and distribution of the time after baseline for the FCb, CRSa, CRSb, and CRSc stool collections?
- What sequence depth was achieved by 16S rRNA sequencing (e.g. median and range of the number of sequences per sample)?
- Were all samples sequenced in one batch or were there multiple batches, in which case were longitudinal samples from the same patient included in the same batch and what adjustments

were performed for batch effects?

- Figs 2-4: It's not made sufficiently clear from the figure legends (or in the associated Results text) that the data shown in the main figures represents just the MM patients rather than the full cohort.
- Fig. 2a, 2d, 2e (also applicable to Supp Fig. 4): the statistical test was mentioned as Wilcoxon rank-sum test, but this is for pairwise comparisons whereas the data presented have five time points (which results in 10 pairwise combinations). How was this addressed? One standard approach for non-parametric testing of repeated measures data would be Friedman's test with post-hoc multiple comparison testing of pairwise combinations.
- Fig 2c: the order of the stacked bars changed, which makes it harder to follow the longitudinal changes in the phyla.
- Fig 2f: this figure panel shows magnitudes of change across time points with an importance score based on a machine learning approach for differentiating time points, but it's not clear which genera had statistically significant shifts across the time points (by one or both of the methods used by the authors). This information is only contained within the Supp Tables. It would be helpful if statistically significant genera were annotated.
- Fig 3b: The p-value is misleading as repeated measurements from the same subjects were included in this plot, inflating significance due to the positive correlation among repeated measurements from the same individual. Repeated measures aware approaches are required, or separate significance calculations for CR vs. PR should be performed at each time point.
- LEfSe analysis shown in Fig 4A and Fig 5A – these figure panels show comparisons of genus and predicted pathway abundances at time points before and after CAR-T infusion. However, these analyses will inflate significance given the treatment of repeated measurements as independent observations.
- Fig 4h – What time point was used for the statistical comparison of pathway abundances shown in this figure panel?
- Fig 5 – how were the cutoffs of $r > 0.2$ and $q < 0.2$ chosen to identify edges to include in this network?

Reviewer #2 (Remarks to the Author):

General review for the authors

The article of Hu et al describes the complex interplay between the gut microbiome and autologous BCMA CAR T-cell therapy in MM, ALL and NHL patients. Using a combination of 16S rRNA gene sequencing, bioinformatics and multiple statistical analysis, this study investigated the temporal changes in the intestinal microbiome during CAR-T cell therapy, the association between microbial communities and clinical response as well as cytokine release syndrome severity. While similar studies were published in the field of checkpoint blockade immunotherapies and allogeneic hematopoietic stem cell transplantation, this study is the first one to decipher the interaction between the gut microbiome and CAR T-cell therapy.

The clinical samples used in this study were obtained from three different patient cohorts, over multiple time points, and were analyzed to comprehensively extract, analyze, and correlate microbiome-based dataset, common biomarkers, immune cells populations, therapeutic outcome and CRS-based adverse events. In that regard, this work could be considered as a resource and the first landmark in the field of immuno-oncomicrobiology associated to CAR T-cells. However, as many papers published recently in the field, this manuscript remains factual, data oriented and lacks mechanistic insights. These insights would be beneficial to specify the mode of action of certain key bacteria or group of bacteria, to propose potential therapeutic interventions and would render the manuscript accessible/appealing to a broader audience. Nevertheless, it is acknowledged the four-way interaction occurring among the gut microbiome, the host immune system, the CAR T-cells and cancer cells is not easy to apprehend, and this task remains extremely complex and daunting.

One of the main findings of this paper was the identification of bacterial genera showing differences in abundance in CR versus PR groups. *Faecalibacterium*, *Bifidobacterium*, *Collinsella* and *Sutterella* were associated with CR. This finding is reminiscent to earlier works on anti-PD-1 checkpoint blockade therapy, suggesting a common effect of such taxa across therapeutic strategies. In addition, by stratifying bacteria genera abundances in two arms, i.e., before and after CAR T-cell therapy, they confidently identified a group of bacteria genera that was enriched

in CR versus PR, in the two arms. This strongly suggests their association to positive clinical outcome of CAR T-cell therapy and put forward their predictive potential. Interestingly, the authors further report that high abundance of genus *Sutterella* was with a prolonged event free survival period following CAR-T therapy. While the authors acknowledged this positive correlation was not systematically observed across different indications, it is a substantial finding that needs to be more extensively discussed and contrasted to former reports to propose a potential mode of action.

The second main finding was the identification of bacterial genera showing differences in abundance in mild versus severe CRS. In particular, *Bifidobacterium* and *Leuconostoc* were found enriched in patient encountering severe CRS. *Bifidobacterium* and *Leuconostoc* negatively correlate with PB monocyte and positively correlated with ferritin/D-dimer proinflammatory molecules, respectively. Again, these are interesting and important findings that unravel a gut microbiome signature associated to CRS and warrant extensive discussion in light of recent literature on CAR T-cell and CRS.

This manuscript should be improved by considering the comments below:

-The recurring question in the field of immune-oncomicrobiology is: is intestinal dysbiosis a cause or a consequence (or both) of immunotherapy. This question remains open in the context of this study. In that regard the authors make a strong statement in the title by using the active form: "Gut microbiome modulates CRS and therapeutic response to CAR T-cell therapy...". Because the cause-and-effect relationship is not proven, this title should be revised to prevent misleading the readers.

-Fig. 1 illustrates the therapeutic outcome of BCMA CAR T-cell therapy and the grade of CRS observed in the three cohorts of patients. However, it does not report any additional treatment given to patients to alleviate their CRS symptoms. Does it mean that tocilizumab was not used in this study? According to CRS management recommendations by Neelapu et al and Lee et al, (doi.org/10.1038/nrclinonc.2017.148, doi.org/10.1182/blood-2014-05-552729, respectively), Tocilizumab should be administered to patients undergoing \geq Grade 2 CRS. It is thus believed that some of the Grade 2-3 CRS events documented in this study were managed by Tocilizumab (and/or other drugs). Please confirm. As this parameter may affect/bias the dataset (cytokines/cell population etc...) obtained at CRS b/c, it should be rigorously documented. This comment holds true for other therapeutics used to blunt inflammation, pathogen, and viral infection.

-If I'm not mistaken, the Simpson index measures population diversity (Fig.2) and the Shannon measures entropy and thus diversity (Fig. 3). Both indexes are being used to assess the diversity of population. Why using both indexes instead of just one in Fig 2 and 3?

-Fig. 2C reports the evolution of the relative abundance of bacterial taxa across therapy stages. The clarity of this plot could be improved by organizing bacterial communities from the highest (top) to the lowest (bottom) abundant one.

-Fig. 3D (and Sup Fig. 7/8) is hard to understand. This may prevent the reader from quickly grasping the take home message. Consider replotting it differently.

-Fig. 4H reports the differential KEGG pathways in CR and PR groups. However, I'm not sure to understand the data processing needed to represent such plot. How do you come up with these different pathways? From the identification of bacteria with 16S rRNA seq? Please clarify and modify the text to ease the comprehension of broad audience readers. This comment holds true for Supplementary Fig. 9 and 10. Regarding these two figures, I'm surprised to see that the arginine and tryptophan metabolism, two pathways commonly associated with CRS, were not identified alongside with the purine/lipoic metabolism and biosynthesis of lipopolysaccharide and peptidoglycan. Could you please comment.

-IL-6, IL-1 α , IL-1 β , M-CSF, MCP-3 and GM-CSF, are key protagonists of CRS (doi.org/10.1038/s41577-021-00547-6). I understand that they are missing from Fig. 5D because they do not fall within the following specs: "Associations with an absolute value of correlation coefficient higher than 0.2 and FDR less than 0.2 were depicted using CytoscapeIs". While I don't

have the proper knowledge to assess the relevance of those specifications, I wonder if they could be adapted to illustrate the correlation between these CRS-related cytokines, the gut microbiome and immune cells? Adding them in the network would be very informative.

On a similar topic, differentiated macrophages play a key role in CRS initiation/mediation. Have you explored the evolution of such immune cell population in your longitudinal analysis? If so, it could be very informative to implement this population in the network of Figure 5D.

Furthermore, Bifidobacterium seems to be increased in severe CRS but at the same time, negatively correlate with monocytes, a major driver of CRS (doi.org/10.1038/s41577-021-00547-6, doi: 10.1038/s41591-018-0041-7, doi.org/10.1038/s41591-018-0036-4). Could you please elaborate on this negative correlation?

-The experimental details regarding the sample preparation, DNA sequencing, data processing, bioinformatics and statistical analysis were thoroughly documented and referenced in the methods section. This section will be very helpful for other teams working in the field, could improve the consistency in the future dataset generated and could allow for better quality meta-analysis. Regarding that last aspect, the data generated in this manuscript (raw and analyzed) must be carefully and comprehensively documented in a source file to ease extraction and utilization of raw data by the scientific community.

-Typo and word inconsistencies were observed throughout the manuscript. This could be sometime misleading (example Line 263, replace decrease by increase). Please thoroughly check the text.

Reviewer #3 (Remarks to the Author):

In this manuscript, Hu and colleagues describe the microbiome composition in patients with MM, ALL, and NHL at multiple timepoints before and during CART cell therapy. The authors reason that the gut microbiome has been shown to impact cancer immunotherapy outcomes, yet such studies of patients undergoing CART cell therapy have been lacking to date. The authors found that certain components of the microbiome were correlated with therapeutic response as well as CART-associated CRS. Overall, this is an important preliminary study of the associations of certain features of the gut microbiome with CART efficacy and toxicities as well as the exploration of potential biomarkers based on microbiome composition to predict clinical outcomes.

The reviewer has major and minor issues with this manuscript:

1. The authors should state the date medians and ranges of each of the five time points (FCa, FCb, CRSa, CRSb, CRSc), as it seems that there could be a wide variety in timing among patients (eg. how many days after CART administration was CRSa taken? Etc). If there is a large variation in these timepoints, the authors should discuss how this may impact results.
2. If there are data available on neurotoxicity, it would be important for the authors to analyze the microbiome in relation to neurotoxicity incidence and severity, as it is a major toxicity associated with CART cell therapy, and there are currently no reliable predictive biomarkers.
3. The authors include a paragraph (lines 239-247) about changes in amino acid metabolism. This could be expanded further and made into its own section, with more details about the impact on microbiome composition on metabolism and how this corresponds to outcomes in immunotherapy.
4. Again, the authors mention changes in amino acid synthesis and metabolism (lines 270-275) and should further explore these observations in relation to CART-associated toxicities.
5. Were there any significant correlations with microbiome composition and CR/PR status or CRS severity when further broken down into subgroups, such as CART cell dose, ASCT status, extramedullary disease, etc?
6. The authors should suggest some potential predictive biomarkers based on their findings and outline future validation strategies for these biomarkers in the discussion section.

Minor comments:

1. Lines 73-76: the authors include different measures for each disease type (eg. 5-year and

median OS for r/r ALL, CR and median OS for DLBCL, and 1-year OS for MM). The authors should provide consistent measures for each disease type for ease of comparison (eg. median OS for ALL, DLBCL, and MM).

2. Lines 226-227: authors mention CR vs PR vs NR for patients with NHL and ALL. Was there no NR group in patients with MM?

3. Some of the sections specify whether associations were found specifically in MM, NHL, or ALL, while other sections do not mention the disease type. The authors should state which findings are significant to which type of malignancy or if the findings are applicable to all three disease types studied (eg. the section "Associations between gut microbiome and CRS" does not state whether these associations were relevant to all cancer types studied, or just patients with MM).

4. Fig. 4B is missing a legend.

Response letter to Reviewers

Response to Reviewer #1 (Remarks to the Author):

The manuscript by Hu et al presents the results of longitudinal fecal 16S rRNA gene sequencing of MM (n=43), B-ALL (n=23), and B-NHL (n=12) subjects receiving CART therapy (5 time points). The authors observed changes in microbiome composition and reduced microbial diversity in MM patients after CART therapy, with similar trends in B-ALL and B-NHL. The authors also reported differences in microbial composition and predicted function across the time points between MM patients with complete response (CR) compared to those with partial response (PR). Microbes including *Sutterella* were reported to be associated with clinical response. Differences in microbial composition and predicted function were also reported between mild and severe cytokine release syndrome (CRS).

This study is motivated by prior research demonstrating an important role for the gut microbiome in modulating immunotherapy response. Extrapolating from this literature, it is probable that the microbiome can be affected by CART therapy and that the microbiome (either baseline profiles or alterations in response to therapy) could influence response to CART therapy. Hu et al use an appropriate strategy to address this important unanswered question: a longitudinal microbiome association study of CART recipients with clinical outcome monitoring. The findings of this manuscript support the concept that the microbiome could be linked to CART response, which would have important implications for the field. However, this study has critical limitations that reduce its impact:

1) Small study size: Most of the results presented in this manuscript are based on the 43 patients with MM. This is a modest size for a microbiome study and raises concerns about the robustness and reproducibility of the reported findings. In particular, the comparisons of complete responders vs. partial responders only utilize 24 and 11 subjects, respectively, and comparisons of grade 1 vs. grade 3 CRS utilized 8 and 19 subjects, respectively. With such small numbers, it is likely that the differential taxa and predictive models shown by the authors are overfitted and won't be reproducible.

Response: Thanks for your comment. While we generally agree with you that the sample size of this study is a modest size, it is still the **largest** study to date to report the relationship between gut microbiome and therapeutic response and cytokine release syndrome in the field of CAR-T cell treatment. More importantly, we have added validation sample with a size of 38 into the study and this makes our study sample size even much large than those reported ones. The following is a list of those reported studies regarding on their sample size and main findings.

Reports	No. of patients	Main findings
---------	-----------------	---------------

	included in the report	
Chaput et al. Ann Oncol. 28: 1368-1379.	26	Faecalibacterium percentages were significantly higher in patients with long-term clinical benefit (n=9), while high proportions of Bacteroides were present in patients with poor clinical benefit (n=17). Microbiota of patients prone to develop colitis (n=7) was enriched in Firmicutes at baseline, while high proportions of Bacteroidetes was observed in metastatic melanoma patients who did not develop colitis after receiving CTLA-4 treatment (n=19).
Dubin et al. Nat Commun. 7: 10391.	34	Taxa within the Bacteroidetes phylum were more prevalent in colitis free patient samples (n=24) compared to the patients who progressed to colitis (n=10) after receiving CTLA-4 therapy.
Gopalakrishnan et al. Science. 359:97-103.	43	Analysis of patient fecal microbiome samples (30 responders, 13 nonresponders) showed significantly higher alpha diversity (p<0.01) and relative abundance of bacteria of the Ruminococcaceae family (p<0.01) in responding patients with melanoma undergoing anti-PD-1 immunotherapy.
Matson et al. Science. 359: 104-108.	42	More abundant Bacterial species in responders (n=16) included Bifidobacterium longum , Collinsella aerofaciens , and Enterococcus faecium . Reconstitution of germ-free mice with fecal material from responding patients resulted in improved tumor control, augmented T cell functions, and greater efficacy of anti-PD-1 therapy.
Holler et al. Biol Blood Marrow Transplant. 20: 640-5.	31	The mean proportion of enterococci in post-transplant stool samples was 21% in patients who did not develop gastrointestinal (GI) graft-versus-host disease (GVHD) compared with 46% in those that subsequently developed GI GVHD and 74% at the time of active GVHD.
Taur et al. Blood. 124: 1174-82.	80	Patients who had a lower diversity of gut microbiota at the time of HSCT had shortened overall survival and higher mortality rates (specifically transplant related mortality), compared with those with a high diversity of gut microbiota.

2) Absence of a validation cohort: Given the great heterogeneity of findings across small microbiome studies, it's important for microbiome association studies to

demonstrate the reproducibility of key findings in an independent validation cohort. This is particularly critical here given the limited number of subjects in the CR vs. PR and CRS grade comparisons. The B-ALL and B-NHL subsets could not be used to validate the CR vs. PR analyses as there were only 2 or 3 subjects with PR, respectively.

Response: As suggested, we validated our main findings in a validation cohort comprised of 38 MM patients, which is a completely independent sample. Consistent with our previous results, we observed decreased overall Shannon diversity (Supplementary Fig. 4C, lines 157-159) and increased abundance of genus *Enterococcus* across the whole therapy (Supplementary Fig. 4H, lines 185-188). In addition, we found that genus *Sutterella* was significantly differentially abundant between the CR and PR group (Supplementary Fig. 5D, lines 243-244).

3) Statistical issues: The study utilizes a longitudinal sample collection strategy but none of the analyses account for the repeated measures design and thereby inflate significance due to positive correlation of the repeated measurements (examples provided under additional comments). Also, the manuscript does not give any indication that p-values were adjusted for multiple hypothesis testing, which is critical given the large number of microbial taxa and predicted pathways that were tested. In addition, none of the analyses were adjusted for covariates (e.g. age, sex, antibiotic use before or during treatment, prior autologous stem cell transplantation, etc.) which could confound the association of microbiota with outcomes.

Response: Considering repeated measurements, we performed Friedman's test for longitudinal analysis of diversity and bacterial taxa, and updated our corresponding Figures 2a, 2c, 2d and 2h. To identify genera and pathways that were associated with the treatment outcomes (PR vs. CR) and CRS grades before and after CAR-T therapy, we applied generalized linear-mixed models (GLMMs) in our analyses which include random effects to account for the within-subject variability. Similarly, we also modified Figures 3a, 3b, 4a, 4b, 4h and Fig. S6. For multiple testing, we did FDR correction for p values of tests identifying differentially abundant microbial taxa and pathways. Further, we also included covariates (i.e., age, sex, number of prior lines of therapy, CAR-T cell dose, Autologous stem cell transplantation, Antibiotic use before or during treatment) into our analyses but we did not observe any obvious differences between different efficacy groups and CRS grade groups. So we did not change our results reported in the main text, but we do provide these new analysis results in Supplement Materials (Supplementary Tables 1 & 2).

4) 16s sequencing with imputed metagenomics was used rather than shotgun metagenomics sequencing: The study would have been strengthened by the use of shotgun metagenomics (as has been used in the prominent immunotherapy microbiome studies), which would allow improved species and possibly strain level

taxonomic assignment as well as more accurate assessment of microbiome functional capacity.

Response: Thanks for your advice. It is true that the shotgun metagenomic sequencing is more informative than the 16S sequencing. As documented in the literature, the 16S sequencing can also produce reliable taxonomic profiling results (Jovel et al., 2016, *Frontiers In Microbiology* 7:459). According to previous report which compared taxonomic profile of the two methods on the same samples (Bokulich et al., 2018, *Microbiome*, 6(1): 1-17. Clooney et al., 2016, *Plos One* 11: e0148028), the two methods yield very comparable results. On the other hand, the 16S appears to be more cost-effective and time saving than the metagenomic sequencing, which makes it more acceptable and advantageous for clinical applications. However, this does not mean we would exclude the shotgun metagenomics sequencing from our research. If possible, we will consider the shotgun metagenomics sequencing approach in our future research.

Additional comments:

- It's not indicated how many samples were collected at each of the five time points from the various subgroups.

Response: We have addressed this point in Supplementary Table 3. A description has been added in the main text (see page 5, lines 109-111).

- What was the median and distribution of the time after baseline for the FCb, CRSa, CRSb, and CRSc stool collections?

Response: Thanks for your helpful suggestion. We have added the relevant part in the revised manuscript (page 6, lines 148-151).

- What sequence depth was achieved by 16S rRNA sequencing (e.g. median and range of the number of sequences per sample)?

Response: We have addressed this point in Supplementary Table 4 and main text (page 5, lines 109-111).

- Were all samples sequenced in one batch or were there multiple batches, in which case were longitudinal samples from the same patient included in the same batch and what adjustments were performed for batch effects?

Response: All samples were sequenced in one batch.

- Figs 2-4: It's not made sufficiently clear from the figure legends (or in the associated Results text) that the data shown in the main figures represents just the MM patients rather than the full cohort.

Response: We have modified the legends for Figures 2-4 to indicate that the results were from MM patients. Thanks for pointing this out.

- Fig. 2a, 2d, 2e (also applicable to Supp Fig. 4): the statistical test was mentioned as Wilcoxon rank-sum test, but this is for pairwise comparisons whereas the data presented have five time points (which results in 10 pairwise combinations). How was this addressed? One standard approach for non-parametric testing of repeated measures data would be Friedman's test with post-hoc multiple comparison testing of pairwise combinations.

Response: Thanks for your suggestion. We have applied Friedman's test for repeated measures data and Wilcoxon rank-sum test for 10 pairwise comparisons of the five timepoints in Figures 2a, 2d, 2e. FDR correction was applied for multiple testing. As a result, we observed significant decrease of Shannon index across the whole treatment period, which provides further support to our previous Wilcoxon rank-sum test results. For comparison of phylum level taxonomy, Firmicutes and Bacteroidetes did not reach significant threshold in Friedman's test, we thus removed these results from our revised manuscript.

- Fig 2c: the order of the stacked bars changed, which makes it harder to follow the longitudinal changes in the phyla.

Response: As suggested, the stacked bars of Figure 2c was ordered. Thanks.

- Fig 2f: this figure panel shows magnitudes of change across time points with an importance score based on a machine learning approach for differentiating time points, but it's not clear which genera had statistically significant shifts across the time points (by one or both of the methods used by the authors). This information is only contained within the Supp Tables. It would be helpful if statistically significant genera were annotated.

Response: As suggested, we have modified the Figure 2f and genera that were identified by both machine learning method and Friedman's test are now bolded and underlined. Also, the legend for this figure was modified to illustrate this point.

- Fig 3b: The p-value is misleading as repeated measurements from the same subjects were included in this plot, inflating significance due to the positive correlation among repeated measurements from the same individual. Repeated measures aware approaches are required, or separate significance calculations for CR vs. PR should be performed at each time point.

Response: As suggested, we performed PcoA analysis for each stage separately. Distance of CRSb stage reached significant ($p = 0.047$).

- LefSe analysis shown in Fig 4A and Fig 5A – these figure panels show comparisons of genus and predicted pathway abundances at time points before and after CAR-T infusion. However, these analyses will inflate significance given the treatment of repeated measurements as independent observations.

Response: Thanks for your advice. We have modified the results of Figures 4A & 5A. For genera that was identified by maSigPro, we applied generalized linear-mixed models (GLMMs) to identify genera differentially abundant before and after CAR-T infusion. Genera with $FDR < 0.05$ are presented in the bar plots of Figure 4A and Supplementary Table 7. Genera that was identified to be differentially abundant by LefSe method are marked with red stars.

- Fig 4h – What time point was used for the statistical comparison of pathway abundances shown in this figure panel?

Response: Figure 4h compared all fecal samples from CR patients with those from PR patients. Considering repeated measurements, we also applied time-course differential analysis (maSigPro) to identify pathways that were differentially abundant between the CR and PR group. Likewise, time-course differential analysis was applied for the differentially abundant pathway analysis concerning CRS grade.

- Fig 5 – how were the cutoffs of $r > 0.2$ and $q < 0.2$ chosen to identify edges to include in this network?

Response: The network in Figure 5 was updated. To better illustrate correlation of CRS-related cytokines with gut microbiome, we additionally checked association of seven CRS-relating cytokines and M1/M2 macrophages with gut microbiome. Moreover, considering the repeated measures design, we applied repeated measures correlation (rmcorr) analysis and updated the network analysis part to show significant correlations of gut microbes with cytokines and immune cells ($FDR < 0.05$).

Response to Reviewer 2 (Remarks to the Author):

General review for the authors

The article of Hu et al describes the complex interplay between the gut microbiome and autologous BCMA CAR T-cell therapy in MM, ALL and NHL patients. Using a combination of 16S rRNA gene sequencing, bioinformatics and multiple statistical analysis, this study investigated the temporal changes in the intestinal microbiome during CAR-T cell therapy, the association between microbial communities and clinical response as well as cytokine release syndrome severity. While similar studies were published in the field of checkpoint blockade immunotherapies and allogeneic

hematopoietic stem cell transplantation, this study is the first one to decipher the interaction between the gut microbiome and CAR T-cell therapy.

The clinical samples used in this study were obtained from three different patient cohorts, over multiple time points, and were analyzed to comprehensively extract, analyze, and correlate microbiome-based dataset, common biomarkers, immune cells populations, therapeutic outcome and CRS-based adverse events. In that regard, this work could be considered as a resource and the first landmark in the field of immunomicrobiology associated to CAR T-cells. However, as many papers published recently in the field, this manuscript remains factual, data oriented and lacks mechanistic insights. These insights would be beneficial to specify the mode of action of certain key bacteria or group of bacteria, to propose potential therapeutic interventions and would render the manuscript accessible/appealing to a broader audience. Nevertheless, it is acknowledged the four-way interaction occurring among the gut microbiome, the host immune system, the CAR T-cells and cancer cells is not easy to apprehend, and this task remains extremely complex and daunting.

One of the main findings of this paper was the identification of bacterial genera showing differences in abundance in CR versus PR groups. *Faecalibacterium*, *Bifidobacterium*, *Collinsella* and *Sutterella* were associated with CR. This finding is reminiscent to earlier works on anti-PD-1 checkpoint blockade therapy, suggesting a common effect of such taxa across therapeutic strategies. In addition, by stratifying bacteria genera abundances in two arms, i.e., before and after CAR T-cell therapy, they confidently identified a group of bacteria genera that was enriched in CR versus PR, in the two arms. This strongly suggests their association to positive clinical outcome of CAR T-cell therapy and put forward their predictive potential. Interestingly, the authors further report that high abundance of genus *Sutterella* was with a prolonged event free survival period following CAR-T therapy. While the authors acknowledged this positive correlation was not systematically observed across different indications, it is a substantial finding that needs to be more extensively discussed and contrasted to former reports to propose a potential mode of action.

The second main finding was the identification of bacterial genera showing differences in abundance in mild versus severe CRS. In particular, *Bifidobacterium* and *Leuconostoc* were found enriched in patient encountering severe CRS. *Bifidobacterium* and *Leuconostoc* negatively correlate with PB monocyte and positively correlated with ferritin/D-dimer proinflammatory molecules, respectively. Again, these are interesting and important findings that unravel a gut microbiome signature associated to CRS and warrant extensive discussion in light of recent literature on CAR T-cell and CRS.

This manuscript should be improved by considering the comments below:

-The recurring question in the field of immune-oncomicrobiology is: is intestinal

dysbiosis a cause or a consequence (or both) of immunotherapy. This question remains open in the context of this study. In that regard the authors make a strong statement in the title by using the active form: “Gut microbiome modulates CRS and therapeutic response to CAR T-cell therapy...”. Because the cause-and-effect relationship is not proven, this title should be revised to prevent misleading the readers.

Response: Thanks very much for your positive comments on our paper and valuable suggestion. As suggested, we have changed our title to “Gut microbiome correlates with cytokine release syndrome and therapeutic response to CAR-T therapy in hematologic malignancies” in the revised manuscript.

-Fig. 1 Illustrates the therapeutic outcome of BCMA CAR T-cell therapy and the grade of CRS observed in the three cohorts of patients. However, it does not report any additional treatment given to patients to alleviate their CRS symptoms. Does it mean that tocilizumab was not used in this study? According to CRS management recommendations by Neelapu et al (Nat Rev Clin Oncol. 2018, 15:47-62) and Lee et al (Blood 2014, 124:188-95), Tocilizumab should be administered to patients undergoing \geq Grade 2 CRS. It is thus believed that some of the Grade 2-3 CRS events documented in this study were managed by Tocilizumab (and/or other drugs). Please confirm. As this parameter may affect/bias the dataset (cytokines/cell population etc...) obtained at CRS b/c, it should be rigorously documented. This comment holds true for other therapeutics used to blunt inflammation, pathogen, and viral infection.

Response: Thanks for your helpful suggestion. We have added this content into the manuscript (see page 5, lines 123-127).

-If I'm not mistaken, the Simpson index measures population diversity (Fig.2) and the Shannon measures entropy and thus diversity (Fig. 3). Both indexes are being used to assess the diversity of population. Why using both indexes instead of just one in Fig 2 and 3?

Response: As suggested, we have changed Simpson index of Figure 2a into Shannon index.

-Fig. 2C reports the evolution of the relative abundance of bacterial taxa across therapy stages. The clarity of this plot could be improved by organizing bacterial communities from the highest (top) to the lowest (bottom) abundant one.

Response: As suggested, the order of taxons in Fig 2c was ordered.

-Fig. 3D (and Sup Fig. 7/8) is hard to understand. This may prevent the reader from quickly grasping the take home message. Consider replotting it differently.

Response: As suggested, we have modified figure 3D, which is a heatmap showing longitudinally differentially abundant OTU clusters between the CR and PR group. Rows are OTUs and columns are fecal samples of subjects in different time points. Heatmap color was proportional to abundance of OTUs, where blue color indicates low abundance and yellow to red color indicate high abundance. Block in left dotted box was abundance and change patterns of OTUs in the three clusters across all the five time points in CR patients. Block in right dotted box was pattern of the three clusters in PR patients. Similarly, we also did changes in Supplementary Figures 7 & 8.

-Fig. 4H reports the differential KEGG pathways in CR and PR groups. However, I'm not sure to understand the data processing needed to represent such plot. How do you come up with these different pathways? From the identification of bacteria with 16S rRNA seq? Please clarify and modify the text to ease the comprehension of broad audience readers. This comment holds true for Supplementary Fig. 9 and 10. Regarding these two figures, I'm surprised to see that the arginine and tryptophan metabolism, two pathways commonly associated with CRS, were not identified alongside with the purine/lipoic metabolism and biosynthesis of lipopolysaccharide and peptidoglycan. Could you please comment.

Response: We applied PICRUSt2 tool to predict functional abundances based on 16S rRNA sequences profiles. PICRUSt2 uses existing annotations of gene content and 16S copy number from reference bacterial genomes in the IMG (Integrated Microbial Genomes) database to predict which gene families are present and then combines gene families to estimate the composite of community KEGG functions. The tool included more than 40,000 bacterial and archaeal genomes from the IMG database and precalculated gene contents for each organism to generate a table of predicted gene family abundances for each organism. Microbial community functions could be inferred by combining the gene content table and relative abundance of 16S rRNA genes in one or more samples (Douglas et al, *Nat Biotechnol*, 2020, 38: 685–688). We have added several sentences to illustrate the method.

As we modified the statistical method to consider repeated measurements, new results for pathway analysis (Figure 4H, Supplementary Figures 9 & 10) were summarized in Supplementary Figures 9 & 10. As the functional pathways of bacteria community were inferred from the bacteria composition, it may not fully represent real bacteria pathways. This bias might lead to omission of some significant pathways such as arginine and tryptophan metabolism pathway. This is considered to be the weakness of 16S rRNA sequencing when comparing with shotgun metagenomic sequencing. To further validate metabolic changes in feces, we applied metabolic Liquid Chromatography Mass Spectrometry (LC-MS) to quantify concentration of fecal metabolites during CRS. The results are summarized in Supplementary Figures 11 & 12. In differential analysis of metabolites between CRS groups, we identified

phosphocreatine which annotated to arginine and proline metabolism to be differentially abundant.

-IL-6, IL-1 α , IL-1 β , M-CSF, MCP-3 and GM-CSF, are key protagonists of CRS (doi.org/10.1038/s41577-021-00547-6). I understand that they are missing from Fig. 5D because they do not fall within the following specs: “Associations with an absolute value of correlation coefficient higher than 0.2 and FDR less than 0.2 were depicted using CytoscapeIs”. While I don’t have the proper knowledge to assess the relevance of those specifications, I wonder if they could be adapted to illustrate the correlation between these CRS-related cytokines, the gut microbiome and immune cells? Adding them in the network would be very informative.

Response: Thanks for your suggestion. To better illustrate correlation of CRS-related cytokines with gut microbiome, we referred to the review summarizing CRS-related cytokines (Li et al., *Signal Transduction and Targeted Therapy*, 2021, 6: 1-16) and added 7 more cytokines (i.e., MIP-1 α , GM-CSF, MCP-1, IL-15, IL-1 β , IL-1 α , IL-17 α) into the network analysis. Moreover, considering the repeated measures design, we applied repeated measures correlation (rmcorr) analysis and updated the network as well. More key protagonists of CRS are presented in the newly generated networks.

On a similar topic, differentiated macrophages play a key role in CRS initiation/mediation. Have you explored the evolution of such immune cell population in your longitudinal analysis? If so, it could be very informative to implement this population in the network of Figure 5D.

Response: We assessed M1 and M2 macrophage by flow cytometry. By associating these two differentiated macrophages with gut microbes, no significant association was for M1 and M2 macrophage after multiple test correction.

Furthermore, Bifidobacterium seems to be increased in severe CRS but at the same time, negatively correlate with monocytes, a major driver of CRS (doi.org/10.1038/s41577-021-00547-6, [doi: 10.1038/s41591-018-0041-7](https://doi.org/10.1038/s41591-018-0041-7), doi.org/10.1038/s41591-018-0036-4). Could you please elaborate on this negative correlation?

Response: No changes are needed as the correlation between Bifidobacterium and monocytes was no longer significant after updating statistic method and multiple test correction.

-The experimental details regarding the sample preparation, DNA sequencing, data processing, bioinformatics and statistical analysis were thoroughly documented and referenced in the methods section. This section will be very helpful for other teams working in the field, could improve the consistency in the future dataset generated and could allow for better quality meta-analysis. Regarding that last aspect, the data

generated in this manuscript (raw and analyzed) must be carefully and comprehensively documented in a source file to ease extraction and utilization of raw data by the scientific community.

Response: As requested by the journal, we will submit all relevant data to the public database as soon as the paper is being accepted by the journal.

-Typo and word inconsistencies were observed throughout the manuscript. This could be sometime misleading (example Line 263, replace decrease by increase). Please thoroughly check the text.

Response: We have double checked the paper. Thanks.

Response to Reviewer 3 (Remarks to the Author):

In this manuscript, Hu and colleagues describe the microbiome composition in patients with MM, ALL, and NHL at multiple timepoints before and during CART cell therapy. The authors reason that the gut microbiome has been shown to impact cancer immunotherapy outcomes, yet such studies of patients undergoing CART cell therapy have been lacking to date. The authors found that certain components of the microbiome were correlated with therapeutic response as well as CART-associated CRS. Overall, this is an important preliminary study of the associations of certain features of the gut microbiome with CART efficacy and toxicities as well as the exploration of potential biomarkers based on microbiome composition to predict clinical outcomes.

The reviewer has major and minor issues with this manuscript:

1. The authors should state the date medians and ranges of each of the five time points (FCa, FCb, CRSa, CRSb, CRSc), as it seems that there could be a wide variety in timing among patients (eg. how many days after CART administration was CRSa taken? Etc). If there is a large variation in these timepoints, the authors should discuss how this may impact results.

Response: Thanks for your kind reminder. We have added these contents into the revised manuscript (see page 6, lines 148-151).

2. If there are data available on neurotoxicity, it would be important for the authors to analyze the microbiome in relation to neurotoxicity incidence and severity, as it is a major toxicity associated with CART cell therapy, and there are currently no reliable predictive biomarkers.

Response: Thanks for your helpful suggestion. In this study, 3 of 43 patients with multiple myeloma (7%) developed *grade 1* neurotoxicity. Our previous study reported

that 5 of 61 MM patients (8.2%) experienced reversible neurotoxicities (Zhang et al, Clin Cancer Res. 27:6384-6392.). Due to the small number of cases (n=3), we were not able to analyze the microbiome in relation to neurotoxicity incidence and severity. Future work with larger samples to explore the relationship between microbiome and neurotoxicity after CAR-T cell treatment.

3. The authors include a paragraph (lines 239-247) about changes in amino acid metabolism. This could be expanded further and made into its own section, with more details about the impact on microbiome composition on metabolism and how this corresponds to outcomes in immunotherapy.

Response: Thanks for your suggestion. we have moved functional analysis into a new section (see page 11, lines 283-303). To better illustrate association of amino acid metabolism with CAR-T therapy, we added Liquid Chromatography Mass Spectrometry (LC-MS) to quantify concentration of fecal metabolites and found several metabolites of amino acid metabolism to be significant different between CRS grade/outcome groups. Moreover, we have added discussions on amino acids and immunotherapy in our revised manuscript (see pages 14-15, lines 387-403).

4. Again, the authors mention changes in amino acid synthesis and metabolism (lines 270-275) and should further explore these observations in relation to CART-associated toxicities.

Response: To further explore metabolites during CRS, we applied Liquid Chromatography Mass Spectrometry (LC-MS) to quantify concentration of fecal metabolites and compared difference between different CRS grade/outcome groups. The results have been updated in the revised ms (see page 11, lines 295-303). In addition, we added sentences in lines 387-403 to discuss amino acids and CART-associated toxicities.

5. Were there any significant correlations with microbiome composition and CR/PR status or CRS severity when further broken down into subgroups, such as CART cell dose, ASCT status, extramedullary disease, etc?

Response: Thanks for your helpful suggestion. We did not break down into subgroups for further analysis because of the limited number of samples in each subgroup. We summarized subgroups information in the following table. For example, there are only 2 PR subjects in non-extramedullary disease subgroup.

	CR N=24(%)	VGPR N=6(%)	PR N=11(%)
CAR-T cell dose($\times 10^6$ /kg)			
Median	4.65	4.03	4.4
Range	1.2-6.9	2.5-6.2	1.3-5.8

Autologous stem cell transplantation			
No	17(70.8)	2(33.3)	5(45.5)
Yes	7(29.2)	4(66.7)	6(54.5)
Extramedullary disease			
No	14(58.3)	2(33.3)	2(18.2)
Yes	10(41.7)	4(66.7)	9(81.8)

6. The authors should suggest some potential predictive biomarkers based on their findings and outline future validation strategies for these biomarkers in the discussion section.

Response: In this paper, we observed significant correlation of gut microbiome with treatment outcome and CRS grade of CAR-T therapy. To further demonstrate the correlation and investigate underline mechanism(s), more wet-lab experiments need to be conducted, which is beyond of the scope of this report. Germ-free mice would be helpful model to validate function of gut microbiome. For example, fecal microbiota transplantation (FMT) from CR and PR patients into germ-free mice to construct mouse colonized by donor microbiota. Then physiological and biochemical response of these FTM-treated mice to myeloma cells and CAR-T cells could be surveyed. Moreover, gut microbiota could be damaged by antibiotic treatment and rescued by FMT, which could help to validate the function of gut microbiome in immunotherapy. On the other hand, genus *Sutteralla* was found to be important biomarkers for therapy outcome. Further study should demonstrate the predictive role of *Sutteralla* in larger cohort of myeloma and explore its predictive capacity in CAR-T therapy of other types of tumors. To study correlation genus *Bifidobacterium* with CRS grade, probiotic supplement of *Bifidobacterium* species to myeloma or CAR-T mouse models could help to reveal immune responses caused by *Bifidobacterium*. Other oral supplement strategies include bacterial metabolites, such as amino acids, fatty acids, cytotoxin, could be used to demonstrate mechanism underlying effect of gut microbes. These points have been added in the revised ms (see page 15, lines 404-421).

Minor comments:

1. Lines 73-76: the authors include different measures for each disease type (eg. 5-year and median OS for r/r ALL, CR and median OS for DLBCL, and 1-year OS for MM). The authors should provide consistent measures for each disease type for ease of comparison (e.g., median OS for ALL, DLBCL, and MM).

Response: Thanks for your suggestions. We have revised our manuscript accordingly (see page 3-4, lines 74-78).

2. Lines 226-227: authors mention CR vs PR vs NR for patients with NHL and ALL. Was there no NR group in patients with MM?

Response: Yes, there was no NR group in patients with MM in this study.

3. Some of the sections specify whether associations were found specifically in MM, NHL, or ALL, while other sections do not mention the disease type. The authors should state which findings are significant to which type of malignancy or if the findings are applicable to all three disease types studied (eg. the section “Associations between gut microbiome and CRS” does not state whether these associations were relevant to all cancer types studied, or just patients with MM).

Response: We have revised the manuscript and specified disease type in both Figure legends (see Figures 1-5) and the main text (see page 6, line 153; page 8, lines 197, 217-218; page 10, line 268).

4. Fig. 4B is missing a legend.

Response: Sorry, the legend for this figure has been added. Thanks for your careful checking our work.

REVIEWER COMMENTS

Reviewer #1 (Remarks to the Author):

The revised manuscript has been significantly improved. The authors have strengthened the conclusions with the addition of a validation cohort of MM patients as well as fecal metabolomics data and addressed some of the concerns about the statistical methodology. The current draft still requires further revision to address remaining statistical questions and better incorporate the new data from the MM validation cohort:

1. The Abstract doesn't currently mention the validation cohort or which findings were validated (in particular, lower alpha diversity and higher *Enterococcus* after CART and higher *Sutterella* in CR vs. PR). This would be important to include in the Abstract to convey the most robust findings from this study given the small sample size.
2. The new LC-MS data is interesting but not currently integrated with the sequencing analyses. Was there functional overlap in differentially abundant metabolites between CR and PR during CRS and differentially abundant predicted pathways? The Results text mentioned phosphonate metabolism but didn't go into this any further or discuss other potentially concordant findings (e.g. phenylalanine metabolism was differentially abundant between CR and PR in Fig. S9 and phenylalanine was increased in CR vs. PR in Fig. S11). Pathway enrichment analysis for the metabolomics data and/or correlation analysis between metabolites and predicted pathway abundances could be considered. The results of such analyses would also inform the Discussion, which currently refers to the functional effects of microbes in a general manner (alluding to amino acid metabolism based on predicted metagenomics).
3. *Actinomyces* was increased after CART in the MLL, NHL, and ALL cohorts. Was this finding validated in the additional 38 MM subjects? These data should be shown regardless as *Actinomyces* is the only genus besides *Enterococcus* that was consistently affected by CART in the three original cohorts.
4. Was Friedman's test for repeated measures used in Fig S4D-G?
5. Was the difference in alpha diversity between PR and CR shown in Fig. 3A validated in the 38 additional MM patients?
6. In Fig 3C-D, was significance of these OTUs determined using a FDR-adjusted threshold as would be appropriate given the large number of features tested? The Figure legend indicates $p < 0.05$. This question also applies to Figs. S7 and S8.
7. Fig 4A appears to show nominal p-values for the maSigPro results. The authors should use FDR-adjusted values (q-values) or provide a line indicating a q-value threshold for significance (e.g. $q < 0.05$) so that it's clear which taxa were significant by maSigPro after adjusting for multiple hypothesis testing. The same comment applies to Fig. 5A, Fig. S9B, and Fig. S10B.
8. In Fig 4B, what was the color code for the three groups shown at each CART timepoint?
9. In Fig. 4C-D, it's unclear what "value" refers to on the y-axis. The Figure legend indicates log transformed relative abundance, but this should result in negative values; $-\log_{10}$ would result in higher values representing lower abundance. This same question applies to Fig 4B. Also, what do the red and blue colors signify (do they represent CR and PR)?
10. Currently the manuscript validates differences in *Sutterella* between CR vs. non-CR in the 38 additional MM subjects (Fig S5D). Was *Sutterella* the only genus that was assessed using this validation cohort? What about the other genera that were suggested by RF analysis as contributing to differentiation of CR vs. PR at baseline and post-chemotherapy (e.g. *Prevotella*, *Collinsella*, *Bifidobacterium*)? Also, why was CR vs. non-CR used instead of CR vs. PR as in the primary analysis?
11. The authors mention multiple genera that were differentially abundant between CR and PR and contributed to RF classifier accuracy. How was *Sutterella* selected for the PFS tertile analysis shown in Fig 4G?
12. In Fig 5B, it's not explained what groups the three colors correspond to (presumably different CRS grades).
13. Did the authors compare *Bifidobacterium* and *Leuconostoc* abundances before and during CRS in the 38 additional MM subjects to validate the results in Fig. 5B?
14. The Results section should clarify whether PICRUSt analysis was only performed for the MM cohort.
15. The PICRUSt findings should be validated using the 38 additional MM subjects to identify which among the many predicted functional shifts were reproducible.

16. Fig. S11 legend needs proofreading ("differentially abundance between the CR and CR groups").
17. The title of Fig. 13S should be changed (it does not show correlation of gut microbes). Also the figure legend should clarify which response groups are represented by the two colors.
18. The Results section (line 323) refers to "multiple inflammatory markers" but lists bacteria; could the authors clarify this?
19. Additional proofreading is required for the new content as multiple typographical errors are present.

Reviewer #2 (Remarks to the Author):

Review of NCOMMS-21-27883A

The initial review from reviewer #2, the author's answers and the comment to author's answers appear in black, blue and red, respectively.

Response to Reviewer 2 (Remarks to the Author):

General review for the authors

The article of Hu et al describes the complex interplay between the gut microbiome and autologous BCMA CAR T-cell therapy in MM, ALL and NHL patients. Using a combination of 16S rRNA gene sequencing, bioinformatics and multiple statistical analysis, this study investigated the temporal changes in the intestinal microbiome during CAR-T cell therapy, the association between microbial communities and clinical response as well as cytokine release syndrome severity. While similar studies were published in the field of checkpoint blockade immunotherapies and allogeneic hematopoietic stem cell transplantation, this study is the first one to decipher the interaction between the gut microbiome and CAR T-cell therapy.

The clinical samples used in this study were obtained from three different patient cohorts, over multiple time points, and were analyzed to comprehensively extract, analyze, and correlate microbiome-based dataset, common biomarkers, immune cells populations, therapeutic outcome and CRS-based adverse events. In that regard, this work could be considered as a resource and the first landmark in the field of immunomicrobiology associated to CAR T-cells. However, as many papers published recently in the field, this manuscript remains factual, data oriented and lacks mechanistic insights. These insights would be beneficial to specify the mode of action of certain key bacteria or group of bacteria, to propose potential therapeutic interventions and would render the manuscript accessible/appealing to a broader audience. Nevertheless, it is acknowledged the four-way interaction occurring among the gut microbiome, the host immune system, the CAR T-cells and cancer cells is not easy to apprehend, and this task remains extremely complex and daunting.

One of the main findings of this paper was the identification of bacterial genera showing differences in abundance in CR versus PR groups. *Faecalibacterium*, *Bifidobacterium*, *Collinsella* and *Sutterella* were associated with CR. This finding is reminiscent to earlier works on anti-PD-1 checkpoint blockade therapy, suggesting a common effect of such taxa across therapeutic strategies. In addition, by stratifying bacteria genera abundances in two arms, i.e., before and after CAR T-cell therapy, they confidently identified a group of bacteria genera that was enriched in CR versus PR, in the two arms. This strongly suggests their association to positive clinical outcome of CAR T-cell therapy and put forward their predictive potential.

Interestingly, the authors further report that high abundance of genus *Sutterella* was with a prolonged event free survival period following CAR-T therapy. While the authors acknowledged this positive correlation was not systematically observed across different indications, it is a substantial finding that needs to be more extensively discussed and contrasted to former reports to propose a potential mode of action.

The second main finding was the identification of bacterial genera showing differences in abundance in mild versus severe CRS. In particular, *Bifidobacterium* and *Leuconostoc* were found enriched in patient encountering severe CRS.

Bifidobacterium and *Leuconostoc* negatively correlate with PB monocyte and positively correlated with ferritin/D-dimer proinflammatory molecules, respectively.

Again, these are interesting and important findings that unravel a gut microbiome signature associated to CRS and warrant extensive discussion in light of recent literature on CAR T-cell and CRS.

This manuscript should be improved by considering the comments below:

-The recurring question in the field of immune-oncomicrobiology is: is intestinal dysbiosis a cause or a consequence (or both) of immunotherapy. This question remains open in the context of this study. In that regard the authors make a strong statement in the title by using the active form: "Gut microbiome modulates CRS and therapeutic response to CAR T-cell therapy...". Because the cause-and-effect relationship is not proven, this title should be revised to prevent misleading the readers.

Response: Thanks very much for your positive comments on our paper and valuable suggestion. As suggested, we have changed our title to "Gut microbiome correlates with cytokine release syndrome and therapeutic response to CAR-T therapy in hematologic malignancies" in the revised manuscript.

Thank you for this important adjustment.

-Fig. 1 Illustrates the therapeutic outcome of BCMA CAR T-cell therapy and the grade of CRS observed in the three cohorts of patients. However, it does not report any additional treatment given to patients to alleviate their CRS symptoms. Does it mean that tocilizumab was not used in this study? According to CRS management recommendations by Neelapu et al (Nat Rev Clin Oncol. 2018, 15:47-62) and Lee et al (Blood 2014, 124:188-95), Tocilizumab should be administered to patients undergoing \geq Grade 2 CRS. It is thus believed that some of the Grade 2-3 CRS events documented in this study were managed by Tocilizumab (and/or other drugs). Please confirm. As this parameter may affect/bias the dataset (cytokines/cell population etc...) obtained at CRS b/c, it should be rigorously documented. This comment holds true for other therapeutics used to blunt inflammation, pathogen, and viral infection.

Response: Thanks for your helpful suggestion. We have added this content into the manuscript (see page 5, lines 123-127).

Thank you for implementing this information into the results section. The use of antibiotics, if any, should also be documented in the text.

How does Toci and corticosteroid affect the results and conclusions drawn out of this study? This point should be discussed.

-If I'm not mistaken, the Simpson index measures population diversity (Fig.2) and the Shannon measures entropy and thus diversity (Fig. 3). Both indexes are being used to assess the diversity of population. Why using both indexes instead of just one in Fig 2 and 3?

Response: As suggested, we have changed Simpson index of Figure 2a into Shannon index.

Please add the statistics between FCa and CRSb cohort as documented in the first version

-Fig. 2C reports the evolution of the relative abundance of bacterial taxa across therapy stages. The clarity of this plot could be improved by organizing bacterial communities from the highest (top) to the lowest (bottom) abundant one.

Response: As suggested, the order of taxons in Fig 2c was ordered.

Thank you. Nothing to add

-Fig. 3D (and Sup Fig. 7/8) is hard to understand. This may prevent the reader from quickly grasping the take home message. Consider replotting it differently.

Response: As suggested, we have modified figure 3D, which is a heatmap showing longitudinally differentially abundant OTU clusters between the CR and PR group. Rows are OTUs and columns are fecal samples of subjects in different time points. Heatmap color was proportional to abundance of OTUs, where blue color indicates low abundance and yellow to red color indicate high abundance. Block in left dotted box was abundance and change patterns of OTUs in the three clusters across all the five time points in CR patients. Block in right dotted box was pattern of the three clusters in PR patients. Similarly, we also did changes in Supplementary Figures 7 & 8.

Thank you. Nothing to add

-Fig. 4H reports the differential KEGG pathways in CR and PR groups. However, I'm not sure to understand the data processing needed to represent such plot. How do you come up with these different pathways? From the identification of bacteria with 16S rRNA seq? Please clarify and modify the text to ease the comprehension of broad audience readers. This comment holds true for Supplementary Fig. 9 and 10.

Regarding these two figures, I'm surprised to see that the arginine and tryptophan metabolism, two pathways commonly associated with CRS, were not identified alongside with the purine/lipoic metabolism and biosynthesis of lipopolysaccharide and peptidoglycan. Could you please comment.

Response: We applied PICRUSt2 tool to predict functional abundances based on 16S rRNA sequences profiles. PICRUSt2 uses existing annotations of gene content and 16S copy number from reference bacterial genomes in the IMG (Integrated Microbial Genomes) database to predict which gene families are present and then combines gene families to estimate the composite of community KEGG functions. The tool included more than 40,000 bacterial and archaeal genomes from the IMG database and precalculated gene contents for each organism to generate a table of predicted gene family abundances for each organism. Microbial community functions could be inferred by combining the gene content table and relative abundance of 16S rRNA genes in one or more samples (Douglas et al, Nat Biotechnol, 2020, 38: 685–688). We have added several sentences to illustrate the method.

As we modified the statistical method to consider repeated measurements, new results for pathway analysis (Figure 4H, Supplementary Figures 9 & 10) were summarized in Supplementary Figures 9 & 10. As the functional pathways of bacteria community were inferred from the bacteria composition, it may not fully represent real bacteria pathways. This bias might lead to omission of some significant pathways such as arginine and tryptophan metabolism pathway. This is considered to be the weakness of 16S rRNA sequencing when comparing with shotgun metagenomic sequencing.

To further validate metabolic changes in feces, we applied metabolic Liquid Chromatography Mass Spectrometry (LC-MS) to quantify concentration of fecal metabolites during CRS. The results are summarized in Supplementary Figures 11 & 12. In differential analysis of metabolites between CRS groups, we identified phosphocreatine which annotated to arginine and proline metabolism to be differentially abundant.

This additional LC-MS dataset is appreciated and helps to consolidate/expand the dataset with an orthogonal technic.

-IL-6, IL-1 α , IL-1 β , M-CSF, MCP-3 and GM-CSF, are key protagonists of CRS (doi.org/10.1038/s41577-021-00547-6). I understand that they are missing from Fig. 5D because they do not fall within the following specs: "Associations with an absolute value of correlation coefficient higher than 0.2 and FDR less than 0.2 were depicted using CytoscapeIs". While I don't have the proper knowledge to assess the relevance of those specifications, I wonder if they could be adapted to illustrate the correlation between these CRS-related cytokines, the gut microbiome and immune cells? Adding them in the network would be very informative.

Response: Thanks for your suggestion. To better illustrate correlation of CRS-related cytokines with gut microbiome, we referred to the review summarizing CRS-related cytokines (Li et al., Signal Transduction and Targeted Therapy, 2021, 6: 1-16) and added 7 more cytokines (i.e., MIP-1 α , GM-CSF, MCP-1, IL-15, IL-1 β , IL-1 α , IL-17 α) into the network analysis. Moreover, considering the repeated measures design, we applied repeated measures correlation (rmcorr) analysis and updated the network as well. More key protagonists of CRS are presented in the newly generated networks. Thank you for considering our suggestion. It was important to present correlation that includes the main protagonists of the CRS.

On a similar topic, differentiated macrophages play a key role in CRS initiation/mediation. Have you explored the evolution of such immune cell population in your longitudinal analysis? If so, it could be very informative to implement this population in the network of Figure 5D.

Response: We assessed M1 and M2 macrophage by flow cytometry. By associating these two differentiated macrophages with gut microbes, no significant association was for M1 and M2 macrophage after multiple test correction.

OK

Furthermore, Bifidobacterium seems to be increased in severe CRS but at the same time, negatively correlate with monocytes, a major driver of CRS (doi.org/10.1038/s41577-021-00547-6, doi: 10.1038/s41591-018-0041-7, doi.org/10.1038/s41591-018-0036-4). Could you please elaborate on this negative correlation?

Response: No changes are needed as the correlation between Bifidobacterium and monocytes was no longer significant after updating statistic method and multiple test correction.

OK

-The experimental details regarding the sample preparation, DNA sequencing, data processing, bioinformatics and statistical analysis were thoroughly documented and referenced in the methods section. This section will be very helpful for other teams working in the field, could improve the consistency in the future dataset generated and could allow for better quality meta-analysis. Regarding that last aspect, the data generated in this manuscript (raw and analyzed) must be carefully and comprehensively documented in a source file to ease extraction and utilization of raw data by the scientific community.

Response: As requested by the journal, we will submit all relevant data to the public database as soon as the paper is being accepted by the journal.

OK

-Typo and word inconsistencies were observed throughout the manuscript. This could be sometime misleading (example Line 263, replace decrease by increase). Please thoroughly check the text.

Response: We have double checked the paper.

The same effort should be done in the revised version as multiple typos remain.

The effort to flesh out the discussion is appreciated. The recently published work of Smith et al (<https://doi.org/10.1038/s41591-022-01702-9>) should be mentioned, contrasted and discuss even though the dataset was acquired from a CD19 CART cell treated cohort of patients.

Reviewer #3 (Remarks to the Author):

The authors have adequately addressed my concerns and comments

Response letter to Reviewers

Response to Reviewer #1 (Remarks to the Author):

The revised manuscript has been significantly improved. The authors have strengthened the conclusions with the addition of a validation cohort of MM patients as well as fecal metabolomics data and addressed some of the concerns about the statistical methodology. The current draft still requires further revision to address remaining statistical questions and better incorporate the new data from the MM validation cohort:

1. The Abstract doesn't currently mention the validation cohort or which findings were validated (in particular, lower alpha diversity and higher Enterococcus after CART and higher Sutterella in CR vs. PR). This would be important to include in the Abstract to convey the most robust findings from this study given the small sample size.

Answer: Thanks for your valuable suggestion. As suggested, we have added the findings from our validation cohort in the Abstract of this revised manuscript.

2. The new LC-MS data is interesting but not currently integrated with the sequencing analyses. Was there functional overlap in differentially abundant metabolites between CR and PR during CRS and differentially abundant predicted pathways? The Results text mentioned phosphonate metabolism but didn't go into this any further or discuss other potentially concordant findings (e.g. phenylalanine metabolism was differentially abundant between CR and PR in Fig. S9 and phenylalanine was increased in CR vs. PR in Fig. S11).

Answer: As suggested, we have compared concordant findings between LC-MS data and PICRUSt-predicted pathways, see lines 337-342 for details.

Pathway enrichment analysis for the metabolomics data and/or correlation analysis between metabolites and predicted pathway abundances could be considered. The results of such analyses would also inform the Discussion, which currently refers to the functional effects of microbes in a general manner (alluding to amino acid metabolism based on predicted metagenomics).

Answer: As suggested, we carried out pathway enrichment analysis for metabolites data which is presented in Supplementary Fig. 15. The results from these new analyses have been incorporated into the revised paper (see lines 342-348 in Results section and lines 444-450 in Discussion section). We further performed correlation analysis between LC-MS-based metabolites and PICRUSt-based predicted pathway abundances during CRS (see the following Fig. 1). However, the results were less conclusive. Considering the two approaches (i.e., pathway enrichment analysis and correlation analysis) suggested by the reviewer serve very similar purpose, we decided to report the results only from the pathway enrichment analysis.

Figure 1: Correlation between LC-MS-based metabolites and PICRUSt-based predicted pathway abundances during CRS in the 38 MM validation cohort. Significance was tested by Spearman correlation analysis. Correlations with FDR < 0.05 were retained. Red and blue color indicate positive and negative correlations, respectively. * FDR < 0.05, ** FDR < 0.01, *** FDR < 0.001.

3. Actinomyces was increased after CART in the MLL, NHL, and ALL cohorts. Was this finding validated in the additional 38 MM subjects? These data should be shown regardless as Actinomyces is the only genus besides Enterococcus that was consistently affected by CART in the three original cohorts.

Answer: We examined the change of Actinomyces during CAR-T therapy in the 38 MM cohorts with Friedman’s test (see Supplementary Fig. 4H) and found the the abundance of Actinomyces increased marginally at CRSc stage comparing with that of FCa and FCb (see line 197; p = 0.064).

4. Was Friedman’s test for repeated measures used in Fig S4D-G?

Answer: For Figure S4D, we used Friedman’s test on subjects who had complete data at all five timepoints (N = 10). The overall significances for Firmicutes and Bacteroidetes were 0.18 and 0.1, respectively (see updated Fig. S4D).

For Figure. S4E, in order to identify genus-level bacteria that was affected by CAR-T therapy in MM patients, we applied both longitudinal analysis (Fig. 2D) and Friedman’s test (Fig. 2E) in our data analyses. Figure S4E showed the trends of significant genera from longitudinal analysis among all 43 patients. The results of Friedman’s test on all patients who had complete data at all five timepoints (N = 10) were presented in Fig. 2E.

For Figures S4F & 4G, we could not apply Friedman’s test because there were only 1 and 2 subjects who had complete data for all five timepoints in the NHL (see below Table 1) and ALL (Table 2) cohort, respectively.

Table 1

subject (NHL)	FCa	FCb	CRSa	CRSb	CRSc
1	Yes	Yes	--	Yes	Yes
2	--	Yes	--	Yes	--
3	Yes	Yes	--	Yes	Yes
4	Yes	Yes	--	Yes	Yes
5	--	--	Yes	--	--
6	--	Yes	--	Yes	Yes
7	Yes	Yes	--	Yes	Yes
8	--	--	--	Yes	Yes
9	Yes	Yes	Yes	Yes	Yes
10	Yes	Yes	--	Yes	Yes
11	--	Yes	--	Yes	Yes
12	Yes	Yes	Yes	--	--

Table2					
Subject (ALL)	FCa	FCb	CRSa	CRSb	CRSc
1	--	Yes	--	Yes	Yes
2	Yes	Yes	--	Yes	Yes
3	--	Yes	Yes	Yes	Yes
4	Yes	Yes	--	Yes	Yes
5	Yes	Yes	--	Yes	Yes
6	Yes	Yes	--	Yes	Yes
7	--	Yes	--	Yes	Yes
8	Yes	Yes	--	Yes	Yes
9	Yes	Yes	--	Yes	Yes
10	--	--	--	Yes	Yes
11	--	Yes	Yes	Yes	Yes
12	Yes	--	--	Yes	Yes
13	--	Yes	--	Yes	Yes
14	--	Yes	--	Yes	Yes
15	--	Yes	--	Yes	Yes
16	Yes	Yes	--	Yes	Yes
17	Yes	Yes	--	Yes	Yes
18	Yes	Yes	Yes	Yes	Yes
19	Yes	Yes	--	Yes	Yes
20	--	Yes	--	Yes	Yes
21	Yes	Yes	--	Yes	Yes
22	Yes	Yes	Yes	Yes	Yes
23	Yes	Yes	--	Yes	Yes

5. Was the difference in alpha diversity between PR and CR shown in Fig. 3A validated in the 38 additional MM patients?

Answer: In the 38 MM validation cohort, Shannon diversity difference between CR and PR group was not significant (see Supplementary Fig. 5G, lines 260-261).

6. In Fig 3C-D, was significance of these OTUs determined using a FDR-adjusted threshold as would be appropriate given the large number of features tested? The Figure legend indicates $p < 0.05$. This question also applies to Figs. S7 and S8.

Answer: For Figure 3D, we summarized number of OTUs with raw p value less than 0.05 given only a limited number of OTU reaches the threshold $FDR < 0.05$. For Figures 3D, S7 and S8, we did use FDR-adjusted threshold. We have updated the corresponding legend for these figures.

7. Fig 4A appears to show nominal p-values for the maSigPro results. The authors should use FDR-adjusted values (q-values) or provide a line indicating a q-value threshold for significance (e.g. $q < 0.05$) so that it's clear which taxa were significant by maSigPro after adjusting for multiple hypothesis testing. The same comment applies to Fig. 5A, Fig. S9B, and Fig. S10B.

Answer: Sorry for confusion. We did consider the issue related to multiple comparisons and all identified genera by maSigPro were indeed significant after correction for multiple testing ($FDR < 0.05$). However, because of our inaccurate labels for x-axis in Fig. 4A and Fig. 5A, it led to confusion and sorry for this. We have updated these figures in the revised manuscript. The labels for Fig. S9B (Fig. S10 in revised version) and Fig. S10B (Fig. S11 in revised version) were right in the previous version.

8. In Fig 4B, what was the color code for the three groups shown at each CART timepoint?

Answer: Different colors represent different response (CR, VGPR and PR). Legend has been added to Fig. 4B. Thanks.

9. In Fig. 4C-D, it's unclear what "value" refers to on the y-axis. The Figure legend indicates log transformed relative abundance, but this should result in negative values; $-\log_{10}$ would result in higher values representing lower abundance. This same question applies to Fig 4B. Also, what do the red and blue colors signify (do they represent CR and PR)?

Answer: The "value" in Fig. 4C-D indicates $\log_2(\text{percentage}+1)$. A pseudo 1 was added to make all the transformed values positive. Red and blue color represent CR and PR, respectively. We have also updated legend for Figure 4.

10. Currently the manuscript validates differences in Sutterella between CR vs. non-CR in the 38 additional MM subjects (Fig S5D). Was Sutterella the only genus that was assessed using this validation cohort? What about the other genera that were suggested by RF analysis as contributing to differentiation of CR vs. PR at baseline and post-chemotherapy (e.g. Prevotella, Collinsella, Bifidobacterium)?

Answer: Thanks for your suggestion. In the discovery MM samples, we revealed a total of four bacteria (i.e., Sutterella, Prevotella, Collinsella and Bifidobacterium) was significantly different between CR and PR groups, with both differential analysis and RF analysis at baseline and post-chemotherapy. Considering the purpose of validation sample, we only wanted to replicate those significant findings in replication sample. Thus, in the 38 validation MM subjects, we validated differences of Prevotella, Collinsella and Bifidobacterium between CR vs. non-CR (Supplementary Figure 5D). These changes have been added into the main text (see lines 261-268).

Also, why was CR vs. non-CR used instead of CR vs. PR as in the primary analysis?

Answer: Given the limited number of PR patients (N = 7) in the 38 validation MM cohort, we merged PR and VGPR (N = 7) group into non-CR group.

11. The authors mention multiple genera that were differentially abundant between CR and PR and contributed to RF classifier accuracy. How was Sutterella selected for the PFS tertile analysis shown in Fig 4G?

Answer: We chose *Sutterella* because it was identified to be differentially abundant between CR and PR at multiple stages by multiple methods (Fig. 4A). Also, it was identified as a top biomarker that discriminated CR from PR by Random Forest procedure at both FCa and FCb stage. In addition, from the boxplot in Fig. S5A, we could see that abundance of Sutterella in CR group was stably higher than that in PR at different stages, indicating that correlation of Sutterella with therapeutic response might be more robust. Thus, we chose Sutterella and checked its association with PFS.

12. In Fig 5B, it's not explained what groups the three colors correspond to (presumably different CRS grades).

Answer: Yes, different colors represent different CRS grades. Figure legend has been added in Fig. 5B.

13. Did the authors compare Bifidobacterium and Leuconostoc abundances before and during CRS in the 38 additional MM subjects to validate the results in Fig. 5B?

Answer: No significance was observed for Bifidobacterium or Leuconostoc in the 38 validation MM patients (see Supplementary Fig. 9, lines 297-299).

14. The Results section should clarify whether PICRUSt analysis was only performed for the MM cohort.

Answer: We have clarified this point as suggested in lines 303 and 307 of the revised manuscript.

15. The PICRUSt findings should be validated using the 38 additional MM subjects to identify which among the many predicted functional shifts were reproducible.

Answer: As suggested, we added differential analysis on PICRUSt predicted pathways of the 38 validation subjects (see Supplementary Fig. 12) and compared with that of the discovery MM cohort. Description of the results were updated in the main text (see lines 313-327).

16. Fig. S11 legend needs proofreading (“differentially abundance between the CR and CR groups”).

Answer: Fig. S11 refers to Fig. S13 in the revised manuscript. The grammar has been corrected.

17. The title of Fig. 13S should be changed (it does not show correlation of gut microbes). Also the figure legend should clarify which response groups are represented by the two colors.

Answer: Fig. S13 refers to Fig. S16 in the revised manuscript. Title and figure legend for this figure have also been updated.

18. The Results section (line 323) refers to “multiple inflammatory markers” but lists bacteria; could the authors clarify this?

Answer: Sorry for the mistake here. What we showed in the paper were indeed those bacteria that correlated significantly with cytokine release syndrome. We have modified the sentence in the revised manuscript (see line 367).

19. Additional proofreading is required for the new content as multiple typographical errors are present.

Answer: Thanks for your kind reminder, we have double checked the paper.

Response to Reviewer #2 (Remarks to the Author):

Review of NCOMMS-21-27883A

The initial review from reviewer #2, the author’s answers and the comment to author’s answers appear in black, blue and red, respectively.

Response to Reviewer 2 (Remarks to the Author):

General review for the authors

The article of Hu et al describes the complex interplay between the gut microbiome and autologous BCMA CAR T-cell therapy in MM, ALL and NHL patients. Using a combination of 16S rRNA gene sequencing, bioinformatics and multiple statistical analysis, this study investigated the temporal changes in the intestinal microbiome during CAR-T cell therapy, the association between microbial communities and clinical response as well as cytokine release syndrome severity. While similar studies were published in the field of checkpoint blockade immunotherapies and allogeneic

hematopoietic stem cell transplantation, this study is the first one to decipher the interaction between the gut microbiome and CAR T-cell therapy.

The clinical samples used in this study were obtained from three different patient cohorts, over multiple time points, and were analyzed to comprehensively extract, analyze, and correlate microbiome-based dataset, common biomarkers, immune cells populations, therapeutic outcome and CRS-based adverse events. In that regard, this work could be considered as a resource and the first landmark in the field of immuno-oncomicrobiology associated to CAR T-cells. However, as many papers published recently in the field, this manuscript remains factual, data oriented and lacks mechanistic insights. These insights would be beneficial to specify the mode of action of certain key bacteria or group of bacteria, to propose potential therapeutic interventions and would render the manuscript accessible/appealing to a broader audience. Nevertheless, it is acknowledged the four-way interaction occurring among the gut microbiome, the host immune system, the CAR T-cells and cancer cells is not easy to apprehend, and this task remains extremely complex and daunting.

One of the main findings of this paper was the identification of bacterial genera showing differences in abundance in CR versus PR groups. *Faecalibacterium*, *Bifidobacterium*, *Collinsella* and *Sutterella* were associated with CR. This finding is reminiscent to earlier works on anti-PD-1 checkpoint blockade therapy, suggesting a common effect of such taxa across therapeutic strategies. In addition, by stratifying bacteria genera abundances in two arms, i.e., before and after CAR T-cell therapy, they confidently identified a group of bacteria genera that was enriched in CR versus PR, in the two arms. This strongly suggests their association to positive clinical outcome of CAR T-cell therapy and put forward their predictive potential.

Interestingly, the authors further report that high abundance of genus *Sutterella* was with a prolonged event free survival period following CAR-T therapy. While the authors acknowledged this positive correlation was not systematically observed across different indications, it is a substantial finding that needs to be more extensively discussed and contrasted to former reports to propose a potential mode of action.

The second main finding was the identification of bacterial genera showing differences in abundance in mild versus severe CRS. In particular, *Bifidobacterium* and *Leuconostoc* were found enriched in patient encountering severe CRS.

Bifidobacterium and *Leuconostoc* negatively correlate with PB monocyte and positively correlated with ferritin/D-dimer proinflammatory molecules, respectively.

Again, these are interesting and important findings that unravel a gut microbiome signature associated to CRS and warrant extensive discussion in light of recent literature on CAR T-cell and CRS.

This manuscript should be improved by considering the comments below:

-The recurring question in the field of immune-oncomicrobiology is: is intestinal dysbiosis a cause or a consequence (or both) of immunotherapy. This question remains open in the context of this study. In that regard the authors make a strong statement in the title by using the active form: "Gut microbiome modulates CRS and therapeutic response to CAR T-cell therapy...". Because the cause-and-effect relationship is not proven, this title should be revised to prevent misleading the

readers.

Response: Thanks very much for your positive comments on our paper and valuable suggestion. As suggested, we have changed our title to “Gut microbiome correlates with cytokine release syndrome and therapeutic response to CAR-T therapy in hematologic malignancies” in the revised manuscript.

Thank you for this important adjustment.

Answer (R2): Thanks for your acknowledgement.

-Fig. 1 Illustrates the therapeutic outcome of BCMA CAR T-cell therapy and the grade of CRS observed in the three cohorts of patients. However, it does not report any additional treatment given to patients to alleviate their CRS symptoms. Does it mean that tocilizumab was not used in this study? According to CRS management recommendations by Neelapu et al (Nat Rev Clin Oncol. 2018, 15:47-62) and Lee et al (Blood 2014, 124:188-95), Tocilizumab should be administered to patients undergoing \geq Grade 2 CRS. It is thus believed that some of the Grade 2-3 CRS events documented in this study were managed by Tocilizumab (and/or other drugs). Please confirm. As this parameter may affect/bias the dataset (cytokines/cell population etc...) obtained at CRS b/c, it should be rigorously documented. This comment holds true for other therapeutics used to blunt inflammation, pathogen, and viral infection.

Response: Thanks for your helpful suggestion. We have added this content into the manuscript (see page 5, lines 123-127).

Thank you for implementing this information into the results section. The use of antibiotics, if any, should also be documented in the text.

How does Toci and corticosteroid affect the results and conclusions drawn out of this study? This point should be discussed.

Answer (R2): Thanks for your helpful suggestion. We have added these contents into the revised manuscript (see page 5, lines 127-134; pages 16, lines 426-429).

-If I'm not mistaken, the Simpson index measures population diversity (Fig.2) and the Shannon measures entropy and thus diversity (Fig. 3). Both indexes are being used to assess the diversity of population. Why using both indexes instead of just one in Fig 2 and 3?

Response: As suggested, we have changed Simpson index of Figure 2a into Shannon index.

Please add the statistics between FCa and CRSb cohort as documented in the first version.

Answer (R2): As suggested, we have added statistics between FCa and CRSb in Figure 2a.

-Fig. 2C reports the evolution of the relative abundance of bacterial taxa across therapy stages. The clarity of this plot could be improved by organizing bacterial communities from the highest (top) to the lowest (bottom) abundant one.

Response: As suggested, the order of taxa in Fig 2c was ordered.

Thank you. Nothing to add

Answer (R2): Thanks for your acknowledgement.

-Fig. 3D (and Sup Fig. 7/8) is hard to understand. This may prevent the reader from quickly grasping the take home message. Consider replotting it differently.

Response: As suggested, we have modified figure 3D, which is a heatmap showing longitudinally differentially abundant OTU clusters between the CR and PR group. Rows are OTUs and columns are fecal samples of subjects in different time points. Heatmap color was proportional to abundance of OTUs, where blue color indicates low abundance and yellow to red color indicate high abundance. Block in left dotted box was abundance and change patterns of OTUs in the three clusters across all the five time points in CR patients. Block in right dotted box was pattern of the three clusters in PR patients. Similarly, we also did changes in Supplementary Figures 7 & 8.

Thank you. Nothing to add

Answer (R2): Thanks for your acknowledgement.

-Fig. 4H reports the differential KEGG pathways in CR and PR groups. However, I'm not sure to understand the data processing needed to represent such plot. How do you come up with these different pathways? From the identification of bacteria with 16S rRNA seq? Please clarify and modify the text to ease the comprehension of broad audience readers. This comment holds true for Supplementary Fig. 9 and 10.

Regarding these two figures, I'm surprised to see that the arginine and tryptophan metabolism, two pathways commonly associated with CRS, were not identified alongside with the purine/lipoic metabolism and biosynthesis of lipopolysaccharide and peptidoglycan. Could you please comment.

Response: We applied PICRUSt2 tool to predict functional abundances based on 16S rRNA sequences profiles. PICRUSt2 uses existing annotations of gene content and 16S copy number from reference bacterial genomes in the IMG (Integrated Microbial Genomes) database to predict which gene families are present and then combines gene families to estimate the composite of community KEGG functions. The tool included more than 40,000 bacterial and archaeal genomes from the IMG database and precalculated gene contents for each organism to generate a table of predicted gene family abundances for each organism. Microbial community functions could be inferred by combining the gene content table and relative abundance of 16S rRNA genes in one or more samples (Douglas et al, Nat Biotechnol, 2020, 38: 685–688). We have added several sentences to illustrate the method.

As we modified the statistical method to consider repeated measurements, new results for pathway analysis (Figure 4H, Supplementary Figures 9 & 10) were summarized in Supplementary Figures 9 & 10. As the functional pathways of bacteria community were inferred from the bacteria composition, it may not fully represent real bacteria

pathways. This bias might lead to omission of some significant pathways such as arginine and tryptophan metabolism pathway. This is considered to be the weakness of 16S rRNA sequencing when comparing with shotgun metagenomic sequencing. To further validate metabolic changes in feces, we applied metabolic Liquid Chromatography Mass Spectrometry (LC-MS) to quantify concentration of fecal metabolites during CRS. The results are summarized in Supplementary Figures 11 & 12. In differential analysis of metabolites between CRS groups, we identified phosphocreatine which annotated to arginine and proline metabolism to be differentially abundant.

This additional LC-MS dataset is appreciated and helps to consolidate/expand the dataset with an orthogonal technic.

Answer (R2): Thanks for your acknowledgement.

-IL-6, IL-1 α , IL-1 β , M-CSF, MCP-3 and GM-CSF, are key protagonists of CRS (doi.org/10.1038/s41577-021-00547-6). I understand that they are missing from Fig. 5D because they do not fall within the following specs: “Associations with an absolute value of correlation coefficient higher than 0.2 and FDR less than 0.2 were depicted using CytoscapeIs”. While I don’t have the proper knowledge to assess the relevance of those specifications, I wonder if they could be adapted to illustrate the correlation between these CRS-related cytokines, the gut microbiome and immune cells? Adding them in the network would be very informative.

Response: Thanks for your suggestion. To better illustrate correlation of CRS-related cytokines with gut microbiome, we referred to the review summarizing CRS-related cytokines (Li et al., Signal Transduction and Targeted Therapy, 2021, 6: 1-16) and added 7 more cytokines (i.e., MIP-1 α , GM-CSF, MCP-1, IL-15, IL-1 β , IL-1 α , IL-17 α) into the network analysis. Moreover, considering the repeated measures design, we applied repeated measures correlation (rmcorr) analysis and updated the network as well. More key protagonists of CRS are presented in the newly generated networks.

Thank you for considering our suggestion. It was important to present correlation that includes the main protagonists of the CRS.

Answer (R2): Thanks for your acknowledgement.

On a similar topic, differentiated macrophages play a key role in CRS initiation/mediation. Have you explored the evolution of such immune cell population in your longitudinal analysis? If so, it could be very informative to implement this population in the network of Figure 5D.

Response: We assessed M1 and M2 macrophage by flow cytometry. By associating these two differentiated macrophages with gut microbes, no significant association was for M1 and M2 macrophage after multiple test correction.

OK

Answer (R2): Thanks for your acknowledgement.

Furthermore, Bifidobacterium seems to be increased in severe CRS but at the same time, negatively correlate with monocytes, a major driver of CRS (doi.org/10.1038/s41577-021-00547-6, doi: 10.1038/s41591-018-0041-7, doi.org/10.1038/s41591-018-0036-4). Could you please elaborate on this negative correlation?

Response: No changes are needed as the correlation between Bifidobacterium and monocytes was no longer significant after updating statistic method and multiple test correction.

OK

Answer (R2): Thanks for your acknowledgement.

-The experimental details regarding the sample preparation, DNA sequencing, data processing, bioinformatics and statistical analysis were thoroughly documented and referenced in the methods section. This section will be very helpful for other teams working in the field, could improve the consistency in the future dataset generated and could allow for better quality meta-analysis. Regarding that last aspect, the data generated in this manuscript (raw and analyzed) must be carefully and comprehensively documented in a source file to ease extraction and utilization of raw data by the scientific community.

Response: As requested by the journal, we will submit all relevant data to the public database as soon as the paper is being accepted by the journal.

OK

Answer (R2): Thanks for your acknowledgement.

-Typo and word inconsistencies were observed throughout the manuscript. This could be sometime misleading (example Line 263, replace decrease by increase). Please thoroughly check the text.

Response: We have double checked the paper.

The same effort should be done in the revised version as multiple typos remain.

Answer (R2): Thanks for your kind reminder, we have double checked the paper.

The effort to flesh out the discussion is appreciated. The recently published work of Smith et al (<https://doi.org/10.1038/s41591-022-01702-9>) should be mentioned, contrasted and discuss even though the dataset was acquired from a CD19 CART cell treated cohort of patients.

Answer (R2): Thanks for your helpful suggestion. We have added this work into the revised manuscript (see page 15, lines 395-401).

REVIEWERS' COMMENTS

Reviewer #1 (Remarks to the Author):

The authors have addressed my concerns and comments in their revised manuscript.

Reviewer #2 (Remarks to the Author):

The authors have addressed all my coments.